# A scalable Li-Al-Cl stratified structure for stable all-solid-state lithium metal batteries

Han Su[1,2], Jingru Li[1,2], Yu Zhong [1] ✉, Yu Liu[1], Xuhong Gao[1], Juner Kuang[1], Minkang Wang[1], Chunxi Lin[1], Xiuli Wang[1] & Jiangping Tu [1] ✉

Sulfides are promising electrolyte materials for all-solid-state Li metal batteries due to their high ionic conductivity and machinability. However, compatibility issues at the negative electrode/sulfide electrolyte interface hinder their practical implementation. Despite previous studies have proposed considerable strategies to improve the negative electrode/sulfide electrolyte interfacial stability, industrial-scale engineering solutions remain elusive. Here, we introduce a scalable Li-Al-Cl stratified structure, formed through the strain-activated separating behavior of thermodynamically unfavorable $Li/Li_9Al_4$ and Li/LiCl interfaces, to stabilize the negative electrode/sulfide electrolyte interface. In the Li-Al-Cl stratified structure, $Li_9Al_4$ and LiCl are enriched at the surface to serve as a robust solid electrolyte interphase and are diluted in bulk by Li metal to construct a skeleton. Enabled by its unique structural characteristic, the Li-Al-Cl stratified structure significantly enhances the stability of negative electrode/sulfide electrolyte interface. This work reports a strain-activated phase separation phenomenon and proposes a practical pathway for negative electrode/sulfide electrolyte interface engineering.

Metallic lithium stands out as the most promising negative electrode material for next-generation, high-energy-density battery technologies, due to its high specific capacity (3860 mAh g⁻¹) and low electrochemical potential (−3.04 V vs. standard hydrogen electrode)[1,2]. Nevertheless, coupling lithium metal with inflammable liquid electrolytes raises a significant safety concern[3]. To address this issue, a viable approach is to construct all-solid-state lithium metal batteries (ASSLMBs) by replacing the liquid electrolytes with solid electrolytes (SEs)[4,5]. Among all the developed SEs, sulfide solid electrolytes (SSEs) have attracted intensive research attention due to their high ionic conductivity and machinability[6,7]. Unfortunately, most SSEs undergo reduction reactions when exposed to the low electrochemical potential of Li metal. This reduction process leads to the formation of an interphase with reduced ionic conductivity, contributing to the buildup of battery resistance. In addition, SSEs suffer from mechanical failure due to Li metal penetration, which ultimately results in short-circuits or micro short-circuits within the battery[8–10].

Considerable efforts have been devoted to enhancing the compatibility between Li metal and SSEs[11–13]. Recently, the artificial construction or in-situ formation of a passivating solid electrolyte interphase (SEI) composed of lithiophobic materials has proven effective in addressing challenges related to SSE reduction and Li penetration[14,15]. However, mechanical failure of the SEI, particularly under high-rate and high-capacity conditions, may arise due to uneven stress distribution caused by inhomogeneous lithium deposition[16,17]. Therefore, additional interface engineering approaches have been proposed to address various concerns associated with this issue. Regarding the insufficient Li⁺ transport kinetics through most passivating SEIs, previous studies have identified a feasible approach involving the introduction of materials with high Li diffusivity to the interphase[18,19]. Besides, due to the weak bonding between lithiophobic materials and metallic Li, inert skeletons are commonly incorporated into the negative electrode structure to avoid void formation during Li stripping[20,21]. In addition, prior works have demonstrated that using a

[1]State Key Laboratory of Silicon and Advanced Semiconductor Materials, Key Laboratory of Advanced Materials and Applications for Batteries of Zhejiang Province, School of Materials Science and Engineering, Zhejiang University, Hangzhou, China. [2]These authors contributed equally: Han Su, Jingru Li. ✉e-mail: yu_zhong@zju.edu.cn; tujp@zju.edu.cn

buffer layer with high mechanical strength can mitigate potential mechanical failures[22,23]. Through these engineering strategies, the prior scenario in which the critical current density (CCD) of ASSLMBs was lower than that of liquid-state lithium-ion batteries has been reversed[24,25]. However, cost-effective pathways for scalable interface construction remain limited. For instance, scaling up techniques such as atomic and molecular layer deposition is challenging[26,27]. Moreover, some designs showcasing superior performance by adding extra interlayers or liquid electrolytes may compromise energy density or safety[28–30]. Thus, there is a compelling need for a practical and scalable negative electrode/SSE interface engineering methodology to meet the transition of ASSLMBs from lab-scale application to industrial implementation.

In this work, we report a practical approach for stabilizing the negative electrode/SSE interface through the use of a Li-Al-Cl stratified structure (LACSS) with a differential surface/bulk phase distribution. This approach involves reactions between nano-$AlCl_3$ precursors and Li metal via a highly scalable sublimating-winding-rolling method. Specifically, due to their significant interfacial repulsion with Li metal, the resulting $Li_9Al_4$ and $LiCl$ products with high specific surface areas will accumulate at the surface and disperse within the bulk alongside unreacted Li metal through strain-activated atom diffusion during the rolling process at room temperature. This unique phase distribution leads to the integration of the SEI and the negative electrode in the LACSS. On the SEI side of the LACSS, a robust passivating SEI with enhanced $Li^+$ transport kinetics is enabled by (1) the sufficiently high Young's modulus of $Li_9Al_4$ and $LiCl$, and (2) the high Li diffusivity of the $Li_9Al_4$ phase and the high lithiophobicity provided by the $LiCl$ phase. On the negative electrode side, the diluted $Li_9Al_4$ and $LiCl$ components serve as an inert skeleton, which bonds the Li metal with the SEI side to prevent void formation. Additionally, $Li_9Al_4$ in the negative electrode side assists in decreasing the Li nucleation potential. Benefiting from these features, the LACSS symmetric cell using $Li_{5.5}PS_{4.5}Cl_{1.5}$ (LPSC) electrolytes exhibits a notable CCD value of $2.5\,\mathrm{mA\,cm^{-2}}$. Besides, under strict conditions with a current density of $1.0\,\mathrm{mA\,cm^{-2}}$ and a capacity of $0.5\,\mathrm{mAh\,cm^{-2}}$, the LACSS symmetric cell can still maintain stable cycling. Furthermore, a remarkable capacity retention of 92.3% is achieved in the LACSS | LPSC | $LiNi_{0.83}Co_{0.12}Mn_{0.05}O_2$ (NCM83125) full cell after 300 cycles at $0.89\,\mathrm{mA\,cm^{-2}}$.

## Results and discussion
### Preparation and characterization of the LACSS
The preparation of the LACSS is realized by chemical reactions between $AlCl_3$ and Li. Figure 1a−e illustrate the schematic diagram of the LACSS preparation, along with the related scanning electron microscopy (SEM)/energy dispersive spectroscopy (EDS) observations

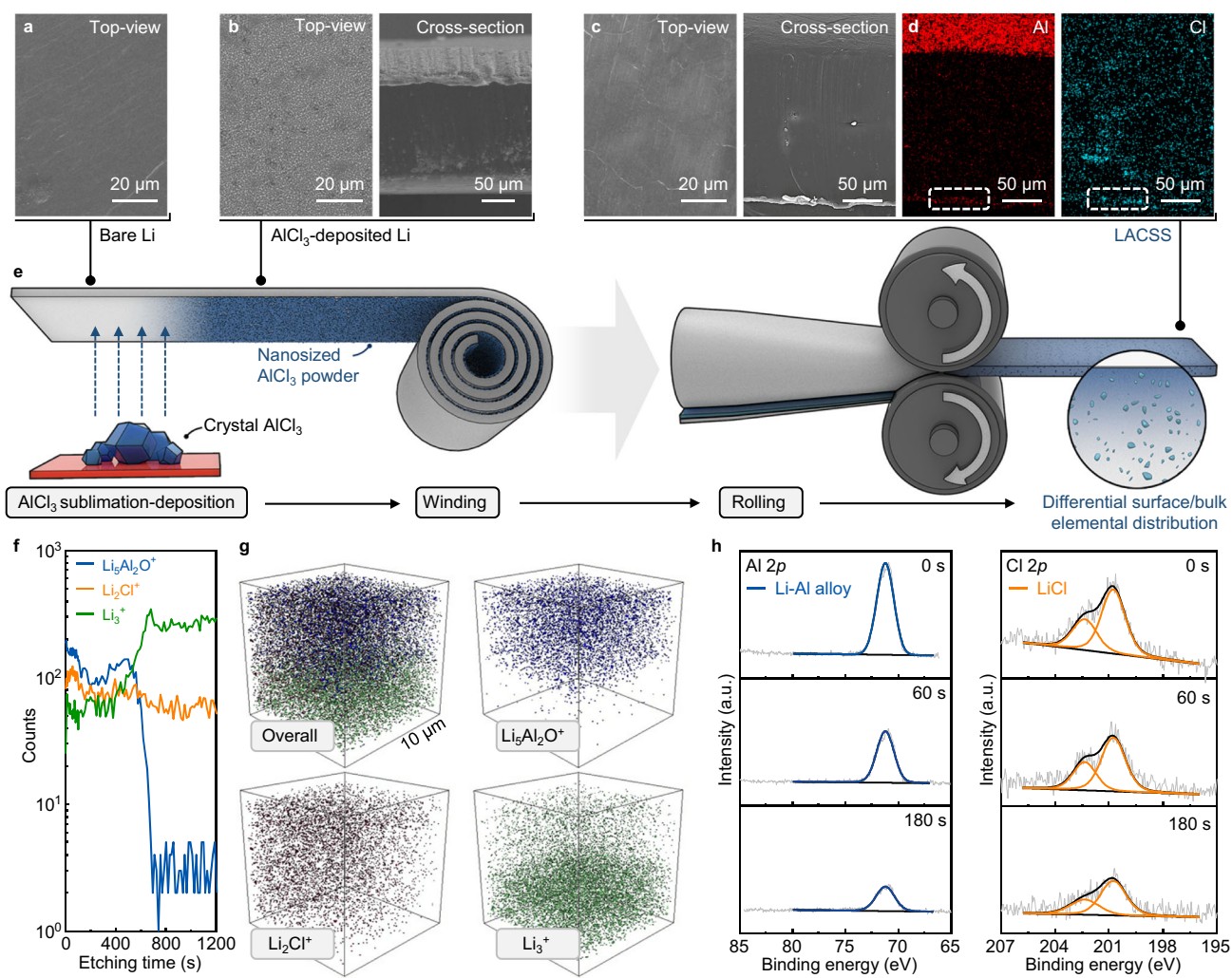

**Fig. 1 | Preparation, structural, and component characterization of the LACSS.** **a** Top-view SEM image of bare Li foil. **b** Top-view and cross-sectional SEM images of the $AlCl_3$-deposited Li foil. **c** Top-view and cross-sectional SEM images of the prepared LACSS, and **d** the corresponding EDS mapping results of Al and Cl. **e** Schematic diagram of the LACSS preparation process. **f** Depth-profiling TOF-SIMS of $Li_5Al_2O^+$, $Li_2Cl^+$, and $Li_3^+$ secondary ion fragments. **g** Overlapping and separate 3-dimensional TOF-SIMS mapping images of $Li_5Al_2O^+$, $Li_2Cl^+$ and $Li_3^+$ signals. **h** In-depth XPS profiles of Al 2$p$ and Cl 2$p$ spectra of the LACSS.

during the preparation. Following 40 s of time-controlled $AlCl_3$ sublimation, a homogenous layer of nanosized $AlCl_3$ powder was deposited on the bare Li foil (Fig. 1a, b and Supplementary Fig. 1). Upon $AlCl_3$ deposition, the surface of the Li foil turned black (Supplementary Fig. 2), indicative of considerable partially reacted $AlCl_3$ powders (Supplementary Fig. 3). Subsequently, a winding-rolling process was employed to facilitate further reaction between the partially reacted $AlCl_3$ and Li metal. SEM image of the $AlCl_3$-deposited Li metal after winding is provided in Supplementary Fig. 4. Further rolling of the $AlCl_3$-deposited Li metal after winding resulted in the formation of the LACSS that displays a differential surface/bulk elemental distribution (Fig. 1c). As depicted by the EDS mapping results of the selected white rectangle in Fig. 1d, the surface of the LACSS exhibits significant enrichment of Al elements and modest enrichment of Cl elements. The production process described above is capable of producing large-sized LACSS for scalable applications (Supplementary Fig. 5). Also, LACSS with different thicknesses can be prepared by adjusting rolling pressure (Supplementary Fig. 6). Furthermore, it should be noted that direct rolling of the $AlCl_3$-deposited Li foils can only lead to a structure comprised of a loose surface and a bulk with few Li-Al and Li-Cl compounds (Supplementary Fig. 7).

Depth-profiling time-of-flight secondary-ion mass spectrometry (TOF-SIMS) in positive mode was further applied to validate the differential surface/bulk elemental distribution in the LACSS. As presented in Fig. 1f, the $Li_5Al_2O^+$ and $Li_2Cl^+$ signals, which represent the Li-Al and Li-Cl reaction products, respectively, exhibit their highest intensities on the surface. The intensity of $Li_5Al_2O^+$ sharply decreases after ~600 s of sputtering, while the intensity of $Li_2Cl^+$ gradually decreases. In contrast to $Li_5Al_2O^+$ and $Li_2Cl^+$ signals, the intensity of the $Li_3^+$ signal maintains the lowest level on the surface. After ~400 s of sputtering, the intensity of $Li_3^+$ endures a rapid increase, which can be ascribed to the presence of unreacted Li metal in the bulk. The three-dimensional overlapping image and separate mapping results of $Li_5Al_2O^+$, $Li_2Cl^+$, and $Li_3^+$ within a $10\,\mu m \times 10\,\mu m$ area are provided in Fig. 1g. In addition, in-depth X-ray photoelectron spectroscopy (XPS) with $Ar^+$ sputtering was applied to characterize the chemical distribution in the LACSS. The total sputtering time was 180 s. As shown in Fig. 1h, consistent peaks corresponding to Li-Al alloys and LiCl are observed in the Al 2$p$ and Cl 2$p$ spectra, respectively, throughout the etching process. Besides, the peak intensities of the Li-Al alloys and LiCl decrease with increasing etching depth, which suggests that Li-Al alloys and LiCl are more concentrated on the surface. In particular, both the XPS and TOF-SIMS results indicate that the concentration difference between the surface and the bulk of Li-Al alloys is more significant than that of LiCl. A possible explanation for this phenomenon is provided in the discussion regarding the formation mechanism of the LACSS.

The phase composition of the LACSS was determined by X-ray diffraction (XRD). When preparing the LACSS for XRD characterization, the exposure time of fresh Li for $AlCl_3$ sublimation was extended to increase the amount of reaction products to the detection limit of the XRD technique. All the peaks in the XRD pattern are indexed to Li, LiCl, and $Li_9Al_4$ (Supplementary Fig. 8), which indicates the complete reduction of $AlCl_3$ by excess Li after the winding-rolling process (Eq. (1)).

$$21\,Li + 4\,AlCl_3 = Li_9Al_4 + 12\,LiCl \qquad (1)$$

Recognizing XRD can only provide an overall result of the surface and bulk phase composition in the structure, a more advanced cryo-focused ion beam scanning electron microscopy (FIB-SEM)/transmission electron microscopy (TEM) integrated technology was applied to analyze the phase distribution in the LACSS. Lamella incorporating the surface and the bulk of the LACSS was prepared via the cryo-FIB-SEM with minimal thermal damage to preserve the native state of the active Li metal-based material. Subsequently, the prepared lamella was transferred to cryo-TEM for later observations under vacuum (Fig. 2a).

The SEM images of key steps during sample preparation are provided in Supplementary Fig. 9. It should be noted that thinning the LACSS sample to a certain extent may lead to its detachment from the protective Pt layer (Supplementary Fig. 10a). Considering that thinning the sample to the maximum extent is crucial for achieving clear observation of lattice fringes, we opt to examine the area where detachment has just started, ensuring minimal sample thickness and maximum preservation of the initial chemical information. As illustrated by the high-angle annular dark-field-transmission electron microscopy (HAADF-TEM) image in Fig. 2b, compared to the bulk region, the surface of the LACSS lamella demonstrates enrichment of heavy atoms as indicated by the increased intensity of imaging spots. These heavy atoms are confirmed to be Al and Cl based on the EDS mappings. According to the line scan profiles of Al and Cl (Fig. 2c), the thickness of the Al/Cl-enriched layer is about 300 nm. In addition, we employed EDS mapping to characterize another LACSS lamella that was relatively thick but had not yet detached from the Pt layer (Supplementary Fig. 10b). As shown in Supplementary Fig. 11, this lamella exhibits an elemental distribution consistent with the region where detachment from the Pt layer has just initiated, confirming the intact preservation of chemical information in our selected region for further TEM characterization.

The selected area electron diffraction (SAED) result of the region in Supplementary Fig. 12 indicates a phase composition of Li, $Li_9Al_4$ and LiCl in the LACSS lamella (Fig. 2d), which is similar to that observed in the XRD pattern. Then, the evolution of phase distribution from the surface region to the bulk region in the LACSS was investigated stepwise by selecting regions 1–3 as highlighted by the squares in Fig. 2e. The fast frontier transformation (FFT) images for regions 1–3 are presented in Fig. 2f–h, respectively. HR-TEM images of regions 1–3 are presented in Supplementary Fig. 13. In the regions near the surface of the LACSS (regions 1 and 2), only the presence of $Li_9Al_4$ and LiCl is identified based on the FFT spots of the LiCl (111) plane along with the $Li_9Al_4$ ((−611), (−101)) planes in region 1 (Fig. 2f) and the FFT spots of LiCl ((200), (111)) planes combined with the $Li_9Al_4$ ((−611), (400), (200), (100)) planes in region 2 (Fig. 2g). When moving further to the bulk (region 3), the FFT probes the appearance of Li metal accompanied by the existence of $Li_9Al_4$ and LiCl (Fig. 2h). These results illustrate that the outermost surface layer of the LACSS, with a thickness of ~100 nm, consists solely of the $Li_9Al_4$ and LiCl phases. Meanwhile, the unreacted Li metal is concentrated in the bulk with diluted $Li_9Al_4$ and LiCl phases.

## Formation mechanism of the LACSS

The distinctive phase distribution in the LACSS implies a tendency for the $Li_9Al_4$ and LiCl phases to separate from the Li matrix during rolling. This phenomenon was also discerned in our investigations of post-reaction products of $AlCl_3$/Li complex prepared via dispersing $AlCl_3$ into molten Li, underscoring the thermodynamic favorability of the separation of $Li_9Al_4$ and LiCl from Li matrix (Supplementary Fig. 14). According to the thermodynamic analysis reported in previous work, the distribution situation of phase X in the matrix is governed by the disparity between the surface energy of phase X ($\gamma_X$) and the phase X/matrix interface energy ($\gamma_{X/M}$)[31]. If $\gamma_X$ is lower than $\gamma_{X/M}$, phase X will separate from the matrix to minimize the overall Gibbs free energy. In addition, a large surface area of phase X will lead to more thermodynamically unfavorable phase X/matrix interfaces, thereby facilitating the separation process. Referring to these factors, the thermodynamic origin of the phase separation in the LACSS can be explained (Fig. 3a). Among the Li-richest alloys, the $Li_9Al_4$ phase (monoclinic) demonstrates a notable structural difference from Li (body-centered cubic). Due to their lattice mismatch, the coupling of Li and $Li_9Al_4$ is likely to present an incoherent interface with a large interface energy, which serves as a thermodynamic driving force for the phase separation of $Li_9Al_4$ and Li. To experimentally validate this hypothesis, we separately introduced Al, In, Sn, and Mg into the molten

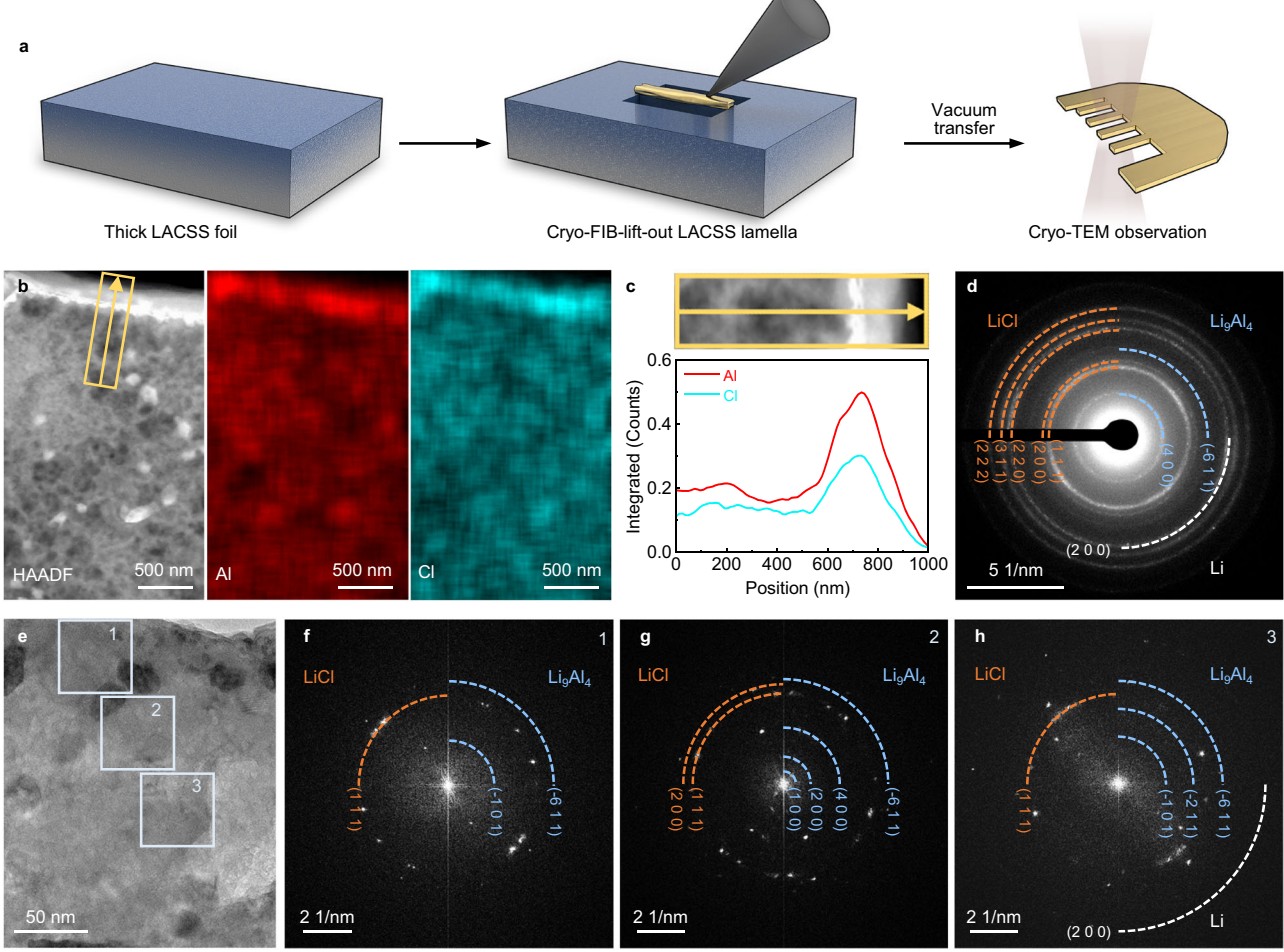

**Fig. 2 | Phase distribution in the LACSS. a** Schematic diagram of sample preparation via the cryo-FIB-SEM. **b** HAADF-TEM image of the LACSS and the corresponding EDS mapping results of Al and Cl. **c** Line scan profiles of the area marked by the yellow rectangle in (**b**). **d** SAED pattern including the surface and the bulk region of the LACSS. **e** HRTEM image of the LACSS where three regions (regions 1–3) are highlighted by blue squares. **f–h** FFT images from region 1 to region 3.

Li metal matrix. The atomic ratio of the incorporated metals (Al, In, Sn, Mg) was controlled to 1.9 at% to ensure an excess of lithium metal. Upon cooling, only $Li_9Al_4$, which exhibits an incoherent interface with the Li matrix, displays phase separation among the prepared mixtures (Supplementary Figs. 15–17). In contrast, the other Li-richest alloys (Li-In, Li-Sn, Li-Mg), possessing a cubic structure similar to that of Li metal, exhibit no noticeable discrepancy between their surface and bulk distributions (Supplementary Fig. 18). As for the LiCl phase, despite its crystal structure similarity to that of Li, phase separation still occurs due to its high lithiophobicity, as revealed by the negative $\gamma_{LiCl} - \gamma_{LiCl/Li}$ value ($-0.42 \, J \, m^{-2}$) in our theoretical study (Supplementary Table 1). Furthermore, considering the relatively high energy barrier of atom diffusion in a solid matrix compared to a molten matrix, a high thermodynamic driving force is necessary for the solid-state system to reach its steady state. In this regard, the high specific surface area of small-sized $Li_9Al_4$ and LiCl prepared via the decomposition of nano-sublimated $AlCl_3$ plays a crucial role in enlarging interfacial repulsions to promote the phase separation process (Fig. 3b). In addition, in line with previous findings, it has been reported that an increased number of exposed surfaces in small-sized particles can enhance atom diffusion across phase boundaries[32]. We verify the size effect through the observation that no visible phase separation occurs between Li and large-sized reaction products when employing micro-sized $AlCl_3$ powders instead of nano sublimated-$AlCl_3$ as reactants (Supplementary Fig. 19).

Based on the above discussion, we propose a mechanism for the LACSS formation as follows. Via the winding process, the partially-reacted $AlCl_3$ precursors are encapsulated in the Li matrix, which increases the reaction area to promote the complete transformation of precursors (Fig. 3c, i). Then, the high strain rate provided by the mechanical rolling process induces the creation of substantial defects (Fig. 3c, ii)[33]. These defects facilitate the diffusion of atoms[34]. During atom diffusion, the partially-reacted nano-$AlCl_3$ undergoes a complete transformation into small-sized $Li_9Al_4$ and LiCl, thereby producing considerable thermodynamically unfavorable $Li/Li_9Al_4$ and $Li/LiCl$ interfaces in the matrix (Fig. 3c, iii). To minimize the overall Gibbs free energy, the ongoing diffusion of atoms leads to the segregation of $Li_9Al_4$ and LiCl from the Li matrix. Consequently, there is a pronounced enrichment of these two phases at the surface of the final structure (Fig. 3c, iv). However, as diffusion progresses, particles of the same phase have the propensity to coalesce into larger entities, hindering their further separation from the Li metal. Notably, due to the higher concentration of LiCl compared to that of $Li_9Al_4$ as indicated by Eq. (1), LiCl is more susceptible to this phenomenon, driven by an increased likelihood of particle encounters. This provides a plausible explanation for the relatively higher proportions of LiCl within the bulk of the LACSS observed from the experimental observations above.

## Stabilization of the negative electrode/SSE interface by the LACSS

The effect of the LACSS on negative electrode/SSE interfacial stability was first elucidated by performing cyclic voltammetry (CV) scans on bare Li and LACSS symmetric cells using LPSC at a potential range of

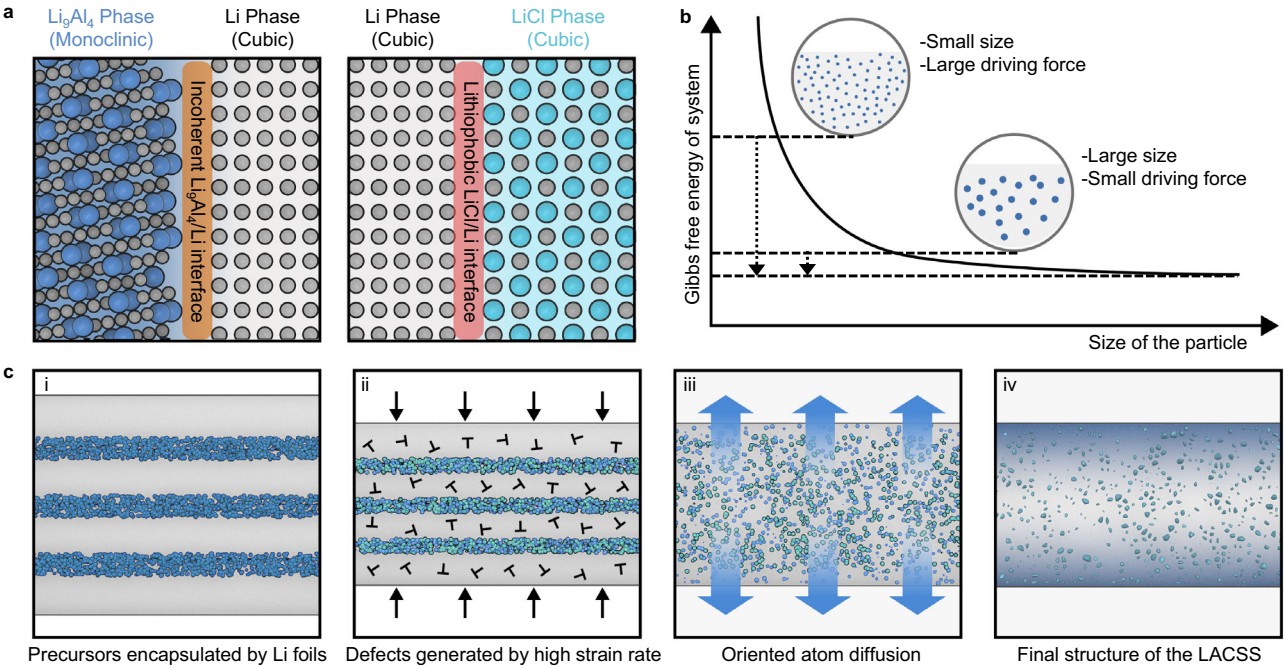

**Fig. 3 | Formation mechanism of the LACSS. a** Thermodynamically unfavorable Li/Li$_9$Al$_4$ and Li/LiCl interfaces. **b** Size effect on phase separation in the solid-state matrix. **c** Schematic diagram of the LACSS formation process.

−0.1 − 0.1 V and a scan rate of 0.2 mV s$^{-1}$. As shown in Fig. 4a, the current of the bare Li symmetric cell significantly increases at the end of the first CV scan, indicating the occurrence of the short-circuit[35]. In contrast, the CV profiles of the LACSS symmetric cell demonstrate high reversibility during 6 scans (Fig. 4b). Then, galvanostatic discharging/charging tests with a step-increased current density of 0.1 mA cm$^{-2}$ were applied to evaluate the CCD of the bare Li and the LACSS symmetric cells (Fig. 4c, d). Compared to the CCD value (1.0 mA cm$^{-2}$) of the bare Li symmetric cell, the CCD value of the LACSS symmetric cell reaches a remarkable level of 2.5 mA cm$^{-2}$. Long-term galvanostatic cycling tests were also conducted on symmetric cells. As presented in Fig. 4e, the bare Li symmetric cell fails after cycling for 160 h at a current density of 0.5 mA cm$^{-2}$ and a capacity of 0.5 mAh cm$^{-2}$. In comparison, the cycle life of the symmetric cell is greatly extended to > 700 h when the LACSS is applied (Fig. 4f). Under more demanding conditions with a current density of 1.0 mA cm$^{-2}$ and a capacity of 0.5 mAh cm$^{-2}$, the LACSS symmetric cell still functions well for more than 400 h while the bare Li symmetric cell encounters short-circuit within 10 h of cycling (Fig. 4g, h). In addition, to investigate the effect of the amount of the reaction precursor AlCl$_3$ on the performance of the final LACSS, we prepared the LACSS with a relatively low AlCl$_3$ loading (L-LACSS) and the LACSS with a relatively high AlCl$_3$ loading (H-LACSS) by adjusting the AlCl$_3$ sublimating time to 20 s and 60 s, respectively. Although L-LACSS and H-LACSS do not outperform LACSS, their symmetric cells exhibit higher CCD values and longer cycle life than the bare Li symmetric cell (Supplementary Fig. 20). In addition, it should be noted that AlCl$_3$-deposited Li processed only by rolling (DRAC-Li) is not applicable for stabilizing the negative electrode/SSE interface, as the symmetric cell using DRAC-Li with a thick and broken surface presents substantially elevated internal resistance compared to its counterpart using LACSS (Supplementary Fig. 21).

Post-mortem analyses were then performed on the bare Li/LPSC and LACSS/LPSC interfaces. As illustrated in Supplementary Fig. 22, the electrochemical impedance spectra (EIS) of the bare Li symmetric cell and the LACSS symmetric cell demonstrate similar resistances before cycling. After 10 cycles of charging/discharging at 0.5 mA cm$^{-2}$/0.5 mAh cm$^{-2}$, the LACSS symmetric cell shows a smaller resistance

increase than the bare Li symmetric cell, indicating that the SEI of the LACSS can decrease side reactions. Moreover, as the linear scanning voltammetry (LSV) profiles in Supplementary Fig. 23 suggest, the exchange current density of the LACSS symmetric cell (1.11 mA cm$^{-2}$) after cycling is almost twice of that of the bare Li symmetric cell (0.69 mA cm$^{-2}$), indicating facilitated Li diffusivity through the SEI of the LACSS. Then, the cycled symmetric cells were disassembled for interface analysis. The XPS results depicted in Supplementary Fig. 24 demonstrate that LPSC and bare Li undergo pronounced reactions after cycling, as evidenced by the considerable amounts of reduction products such as Li$_2$S and reduced P at the interface. The reduction of LPSC is notably alleviated when the bare Li is replaced by the LACSS. As illustrated by the XPS results of the LACSS/LPSC interface (Supplementary Fig. 25), only a minor quantity of reduction products formed through the reaction between Li$_9$Al$_4$ and LPSC is observed. The microscopic characterization was further applied to study the cycled interfaces. As shown in the cross-sectional SEM images of the bare Li/LPSC and LACSS/LPSC interfaces, after 10 cycles of Li stripping/plating, the in-situ formed SEI in the bare Li symmetric cell becomes incapable of impeding lithium penetration into the LPSC, resulting in severe structural damage (Fig. 5a). In sharp contrast, with the intervention of the LACSS, mechanical failure of the SSE is effectively mitigated, as revealed by the structural integrity of the LPSC. In addition, the EDS mapping results reveal that Al and Cl remain enriched on the surface of the LACSS while being diluted in the bulk (Fig. 5b), indicating the preservation of the LACSS's unique structure during the electrochemical process. Furthermore, to investigate the influence of LACSS's distinctive structural characteristics on Li plating behavior, we deposited 2 mAh cm$^{-2}$ of metallic Li onto the LACSS electrode at a current density of 0.1 mA cm$^{-2}$. Even following substantial lithium deposition, the enrichment of Al and Cl persists on the surface of the LACSS, as evidenced by the EDS mapping results. In addition, no intrusion of metallic lithium into the LPSC is observed (Supplementary Fig. 26). These findings suggest that Li deposition occurs beneath the Al/Cl-enriched layer of the LACSS. Referring to previous reports of the Li alloy/LiCl hybrid SEI[18], a potential explanation for this phenomenon is that despite the presence of Li$_9$Al$_4$ with a

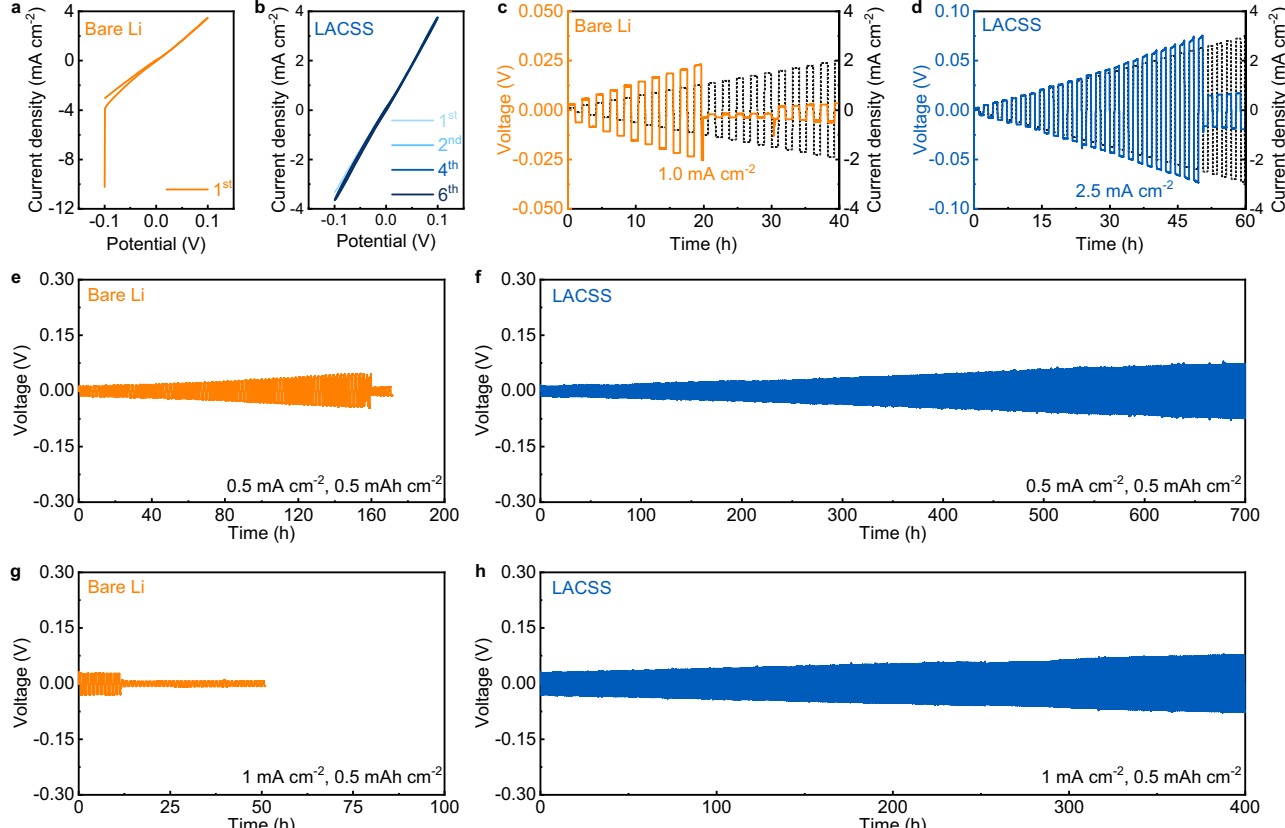

**Fig. 4 | Electrochemical performance of the bare Li and the LACSS symmetric cells. a, b** CV curves of the bare Li symmetric cell (**a**), and the LACSS symmetric cell (**b**). **c, d** Voltage profiles of the bare Li symmetric cell (**c**) and the LACSS symmetric cell (**d**) during galvanostatic discharging/charging tests at step-increased current densities. **e, f** Galvanostatic cycling performance of the bare Li symmetric cell (**e**) and the LACSS symmetric cell (**f**) at 0.5 mA cm$^{-2}$/0.5 mAh cm$^{-2}$. **g, h** Galvanostatic cycling performance of the bare Li symmetric cell (**g**) and the LACSS symmetric cell (**h**) at 1 mA cm$^{-2}$/0.5 mAh cm$^{-2}$.

certain electronic conductivity, the enriched LiCl on the outermost layer of LACSS can decrease the overall electronic conductivity, thereby facilitating Li deposition underneath the $Li_9Al_4$/LiCl-enriched layer. Therefore, we propose that the differential distribution of surface/bulk phases in the LACSS achieves an integrated SEI/negative electrode feature. The enriched $Li_9Al_4$ and LiCl on the surface comprise the SEI, while the unreacted lithium metal, accompanied by diluted $Li_9Al_4$ and LiCl, constitutes the negative electrode.

The enhanced negative electrode/SSE stability can be ascribed to the following physiochemical merits of the LACSS. Regarding the SEI aspect of the LACSS, $Li_9Al_4$ and LiCl, with a high theoretical Young's modulus of 62.4 GPa and 49.9 GPa[36,37], respectively, contribute to elevating the mechanical strength to suppress mechanical failure. As observed by atomic force microscopy (AFM), the dense surface of the LACSS exhibits an extremely high average Young's modulus of 64 GPa, which plays a significant role in depressing potential Li penetration (Fig. 5c–d). Besides, the $Li_9Al_4$/LiCl-enriched SEI diminishes the reduction of LPSC. LiCl maintains thermodynamic stability with the LPSC, while $Li_9Al_4$ exhibits better stability towards SSE compared to metallic Li. This is evidenced by the structural variations observed in the Li/LPSC and $Li_9Al_4$/LPSC interface models during ab initio molecular dynamics (AIMD) simulations. Exposure of LPSC to metallic lithium results in severe decomposition of the $PS_4^{3-}$ tetrahedra, as depicted in screenshots of the Li/LPSC interface before and after AIMD simulations, as well as the corresponding radial distribution function (RDF) profile of the P-S pair (Fig. 5e). In contrast, fewer decompositions of the $PS_4^{3-}$ tetrahedra are observed when the LPSC is coupled with $Li_9Al_4$ (Fig. 5f). Besides, the high Li diffusivity of $Li_9Al_4$ and the short Li$^+$ diffusion distance provided by the thin interlayer accelerate Li$^+$

transport through the SEI of the LACSS[38]. Concerning the negative electrode side of the LACSS, the dispersed $Li_9Al_4$ and LiCl act as frameworks, establishing a bond between the negative electrode and the SEI to inhibit the formation of voids. In addition, according to our theoretical investigation, the Li atom adsorption energy of $Li_9Al_4$ (−1.97 eV) surpasses that of metallic lithium (−1.52 eV) (Supplementary Fig. 27), suggesting reduced resistance for Li nucleation on $Li_9Al_4$ sites[39].

## Performance of ASSLMBs using the LACSS

Using LPSC as the interlayer, ASSLMBs paired with a bare Li or LACSS negative electrode and a high-voltage NCM83125 positive electrode were constructed. The long-term cycling performance of ASSLMBs was first examined at 30 °C. As shown in Fig. 6a, the LACSS|LPSC| NCM83125 full cell delivers a maximum areal capacity of 1.30 mAh cm$^{-2}$ at 0.89 mA cm$^{-2}$. After 300 cycles, it maintains a reversible areal capacity of 1.20 mAh cm$^{-2}$, with a capacity retention of 92.3%. This remarkable performance is primarily attributed to the enhanced negative electrode/SSE interfacial stability, as indicated by the consistently low voltage polarization throughout the cycles of the full cell using the LACSS (Fig. 6b). In contrast, when the LACSS is replaced by bare Li, the full cell experiences substantial capacity decay due to severe voltage polarization and encounters a short-circuit at the 80$^{th}$ cycle (Fig. 6c).

The effect of the LACSS on full cells' rate performance was also evaluated. In comparison with the inferior rate performance represented by the Li|LPSC|NCM83125 full cell, the full cell using the LACSS demonstrates notable rate performance, achieving capacities of 200.9, 172.9, 161.6, 147.5, 129.7, and 97.2 mAh g$^{-1}$ at 0.05, 0.1, 0.2, 0.5, 1, and 2 C

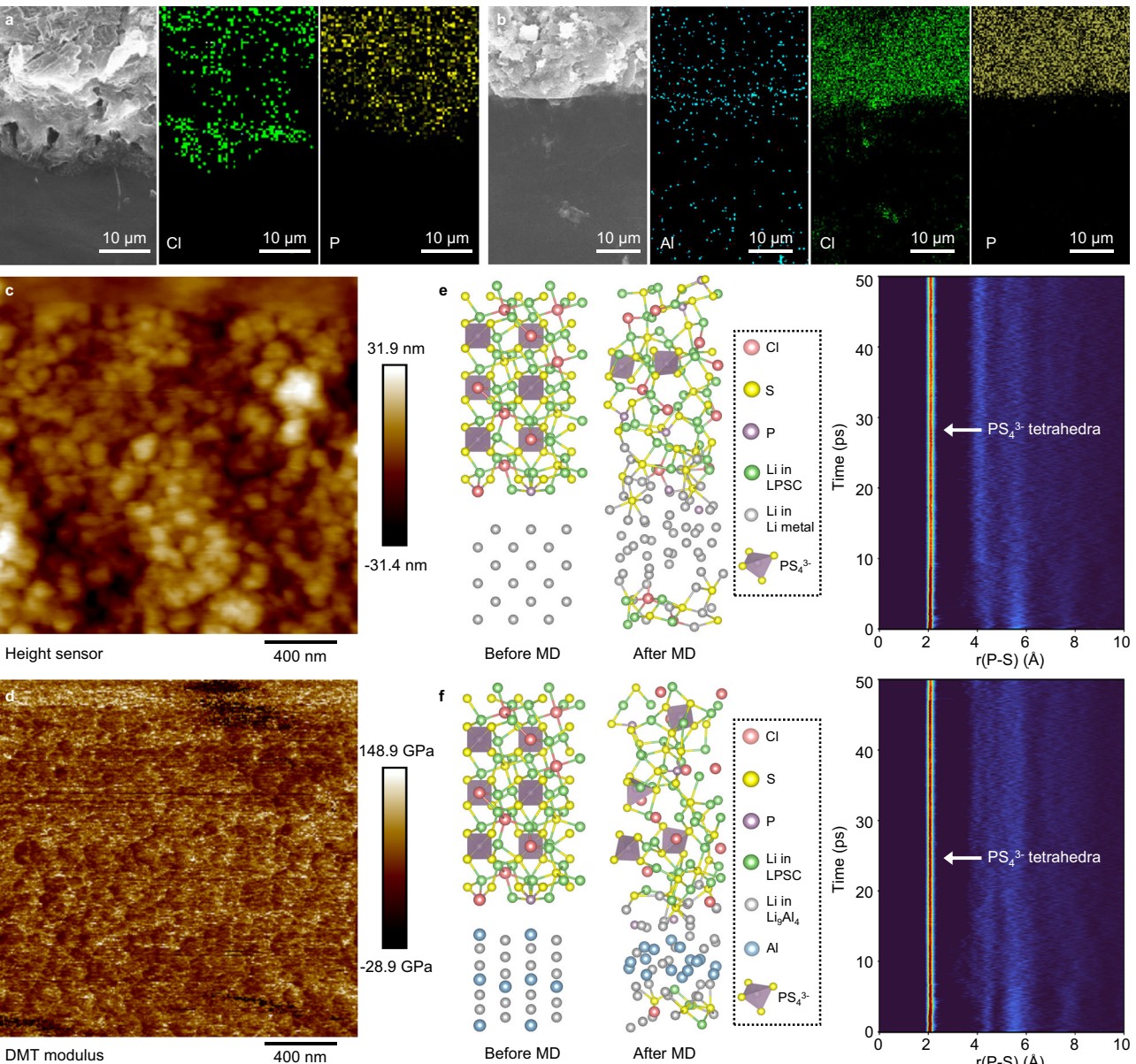

**Fig. 5 | Stabilization of the negative electrode/SSE interface by the LACSS.**
**a** Cross-sectional SEM image of the bare Li/LPSC interface and corresponding EDS mapping results of Cl and P. **b** Cross-sectional SEM image of the LACSS/LPSC interface and corresponding EDS mapping results of Al, Cl and P. **c**, **d** AFM signals of height (**c**) and modulus (**d**) on the surface of the LACSS. **e**, **f** Snapshots of the Li/LPSC interface (**e**) and $Li_9Al_4$/LPSC interface (**f**) before and after AIMD simulations and the corresponding RDF profiles of the P-S pair.

$(1 C = 1.07 \, mA \, cm^{-2})$, respectively. After the current density is decreased to 0.1 C, the LACSS | LPSC | NCM83125 full cell still showcases a high discharging capacity of 162.8 mAh $g^{-1}$, which remains at 155.3 mAh $g^{-1}$ after the following 70 cycles (Fig. 6d). Furthermore, galvanostatic intermittent titration technique (GITT) tests were conducted on the LACSS | LPSC | NCM83125 and Li | LPSC | NCM83125 full cells after 5 cycles at 0.05 C $(1 C = 1.07 \, mA \, cm^{-2})$. As shown in Fig. 6e, f, the LACSS | LPSC | NCM83125 full cell presents a lower voltage polarization, signifying that the application of the LACSS effectively decreases the side reactions.

In summary, we construct a SEI/negative electrode integrated LACSS via a highly scalable sublimating-winding-rolling process for practical negative electrode/SSE interface engineering. In the LACSS, $Li_9Al_4$ and LiCl are enriched at the surface to serve the SEI and are diluted in bulk by Li metal to construct a skeleton in the negative electrode. This unique phase distribution is achieved through the

thermodynamically favorable separation of $Li_9Al_4$ and LiCl products with high specific surface areas from the Li matrix when a high strain rate is provided by the rolling process. The LACSS effectively stabilizes the negative electrode/SSE interface. The LACSS symmetric cell yields a remarkable CCD value of 2.5 mA $cm^{-2}$. Additionally, under stringent conditions with a current density of 1.0 mA $cm^{-2}$ and a capacity of 0.5 mAh $cm^{-2}$, the LACSS symmetric cell exhibits consistent cycling stability for 400 h. Moreover, when coupled with the NCM83125 positive electrode, the LACSS enables the ASSLMB to achieve stable cycling for 300 cycles with a superior capacity retention of 92.3%. The superior performance of the LACSS is provided by its passivating SEI side with high Li diffusivity and mechanical strength, as well as its negative electrode side with inert skeletons and decreased Li nucleation potential. The notable achievement of the scalable LACSS in stabilizing the negative electrode/SSE interface sheds light on industrial production. Beyond this point, the strain-activated separation of

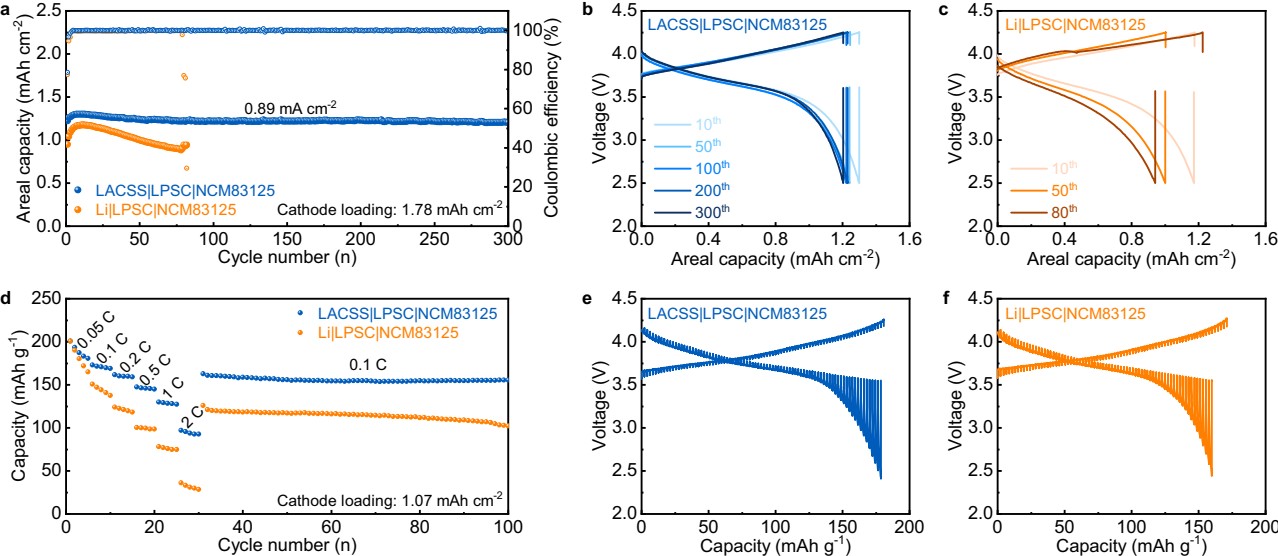

**Fig. 6 | Electrochemical performance of full cells with bare Li or LACSS. a** Cycling performance of the full cells using bare Li or LACSS at 0.89 mA cm⁻². **b, c** Voltage profiles of the full cell using LACSS as the negative electrode (**b**) and the full cell using bare Li as the negative electrode (**c**). **d** Rate performance of full cells at various rates from 0.05 C to 2 C. **e, f** GITT tests of the full cell using LACSS as the negative electrode (**e**) and the full cell using bare Li as the negative electrode (**f**) after 5 cycles at 0.05 C.

thermodynamically unfavorable interfaces reported in this study may inspire future explorations on functional material designs.

## Methods

### Synthesis of the LACSS

Lithium (Li) foil (99.95%, China Energy Lithium Co., Ltd) was first polished by a plastic scraper to remove the native oxidation layer. Then, the polished Li foil was exposed to the gas-state $AlCl_3$ (99.99%, Macklin) at a concentration of 3.6 g L⁻¹ at 178 °C, generating a layer of $AlCl_3$ deposits on the surface. The amount of deposited $AlCl_3$ was adjusted by controlling the exposure time. After cooling, a winding-rolling process was conducted at room temperature to obtain the final LACSS.

### Cell assembly and electrochemical measurements

The LPSC electrolytes used for electrochemical tests were prepared via the following approach[40]. Firstly, stoichiometric $Li_2S$ (99.98 %, Sigma-Aldrich), $P_2S_5$ (99 %, Macklin), and LiCl (99.9 %, Aladdin) were first mechanically milled at 550 rpm for 20 h. Then, as-milled precursors obtained from the ball milling process were further annealed at 450 °C for 5 h to yield the final LPSC products. The XRD pattern of the synthesized LPSC is provided in Supplementary Fig. 28a. The ionic conductivity of the synthesized LPSC was measured to be 7.9 mS cm⁻¹ (Supplementary Fig. 28b).

All the solid-state cells were assembled in the Ar-filled glovebox ($H_2O$ < 0.1 ppm, $O_2$ < 0.1 ppm). For symmetric cells, electrolyte pellets were prepared by cold pressing LPSC powder (150 mg) at 360 MPa inside the poly (ether-ether-ketone) (PEEK) mold with a diameter of 10 mm. EIS measurements were carried out on a Princeton PARSTAT MC-1000 potentiostat in the frequency range of 1 M - 0.1 Hz. CCD and galvanostatic Li stripping/plating measurements were conducted on the Neware CT-4008 battery test system. CV and LSV tests were performed on the CHI 760E electrochemical workstation at a voltage range of − 0.1 V − 0.1 V. The scan rates for CV tests and LSV tests were 0.2 mV s⁻¹ and 1 mV s⁻¹, respectively. All the electrochemical tests on Li symmetric cells were performed in the thermostatic chamber at a temperature of 25 °C.

Full cells using Li metal or LACSS as the negative electrode were assembled as follows. Firstly, a composite positive electrode was prepared by mixing Zr/F doped NCM83125 (purchased from Ningbo Ronbay New Energy Technology Co. Ltd.) and LPSC with a weight ratio

of 70:30 together at 110 rpm for 1 h via Retsch PM-100[41]. Then, 120 mg of LPSC powder was pressed at 240 MPa for 1 min inside the PEEK mold with a diameter of 10 mm to form the electrolyte pellet. After that, the composite positive electrode was spread on the electrolyte pellet uniformly. The bilayer composite positive electrode/LPSC pellet structure was pressed under 360 MPa for 1 min. Finally, 100 μm-thick bare Li or 60 μm-thick LACSS was attached to the other side of the LPSC pellet. All the full cells were tested under a pressure of 6 MPa (Supplementary Fig. 29). The long-term cycling performance, rate performance, and GITT tests of full cells were conducted via the NEWARE CT-4008 battery test system within the voltage range of 2.5-4.25 V. All the electrochemical tests on full cells were performed in the thermostatic chamber with a temperature of 30 °C.

Specifically, the areal capacity of the 60 μm-thick LACSS that applied in the full cell was measured to be 11 mAh cm⁻² by discharging the LACSS || Cu half-cell using a liquid electrolyte (1 M $LiPF_6$ in EC: DEC (1: 1 in volume) with 5 wt% FEC) to the cut-off voltage of −0.5 V (Supplementary Fig. 30). The areal capacity of the 100 μm-thick bare Li was calculated to be 20 mAh cm⁻². The positive electrode loading of the full cells for long-term cycling performance tests was 1.78 mAh cm⁻². The theoretical N/P ratios of the LACSS | LPSC | NCM83125 full cell and the Li | LPSC | NCM83125 used in the long-term cycling stability tests were 6.18 and 11.23, respectively. The positive electrode loading of the full cells for rate and GITT tests was 1.07 mAh cm⁻². The theoretical N/P ratios of the LACSS | LPSC | NCM83125 full cell and the Li | LPSC | NCM83125 used in rate performance and GITT tests were 10.28 and 18.69, respectively. GITT tests were performed by charging/discharging the cell at 0.05 C (1 C = 1.07 mAh cm⁻²). Each step during the GITT test consisted of a galvanostatic current pulse for 5 min and a subsequent rest period for 25 min.

### Characterizations

X-ray diffraction (XRD) was performed on the Rigaku SmartLab instrument with copper Kα radiation at a scan rate of 10° min⁻¹. Scanning electron microscope (SEM) images and energy dispersive spectroscopy (EDS) were collected by Hitachi S4800. X-ray photoelectron spectroscopy (XPS) was conducted on a Thermo Scientific K-Alpha+ system with an Al-Kα source. The time-of-flight secondary ion mass spectrometry (TOF-SIMS) was performed on the IONTOF GmbH 5-100 instrument under the positive ion mode. A focused ion beam scanning

electron microscope (FIB-SEM, Helios 5 UX, ThermoFisher SCIENTIFIC) with a cryo-stage was used to fabricated the lamella of the LACSS. Then, transmission electron microscopy (TEM) was conducted on Talos F200X G2 (ThermoFisher SCIENTIFIC) with a cryo-stage. All the samples were transferred to the sample holder within the Ar atmosphere.

## Theoretical calculations

All theoretical studies were achieved through first-principles calculations using the Vienna Ab initio Simulation Package 5.4.4 (VASP5.4.4). The ion-electron interaction was represented employing the projector augmented wave method[42], while the calculation of exchange-correlation energy utilized the Perdew-Burke-Ernzerhof (PBE) functional within the generalized gradient approximation[43]. All geometry optimizations were performed using the conjugated gradient method, and the convergence criteria were set to $10^{-4}$ eV for energy and 0.01 eV Å$^{-1}$ for force. A K-point mesh of $3 \times 1 \times 1$ and a cut-off energy of 400 eV were used to calculate the interface energy and surface energy, and a K-point mesh of $12 \times 12 \times 12$ and a cut-off energy of 600 eV were used to calculate the bulk energy of Li and LiCl. The surface energy of LiCl and the interface energy between Li and LiCl were calculated as follows[23]. In detail, the surface energy ($\gamma$) of the LiCl (100) surface was calculated using Eq. (2), where $A$ is the surface/interface area, $E_{\text{slab}}$ is the total energy of the LiCl(100) slab, and $E_{\text{bulk}}$ is the bulk energy of LiCl.

$$\gamma = \frac{1}{2A}\left(E_{\text{slab}} - E_{\text{bulk}}\right) \quad (2)$$

The calculation details of the interface energy were listed as follows.

$$2A\gamma\left(\text{Li/LiCl}\right) = E\left(\text{Li/LiCl}\right) - n_{\text{Li}}E\left(\text{Li}\right) - n_{\text{LiCl}}E\left(\text{LiCl}\right) \quad (3)$$

$$\gamma\left(\text{Li/LiCl}\right) = \gamma^{\text{True}}\left(\text{Li/LiCl}\right) + n_{\text{Li}}\sigma \quad (4)$$

In Eq. (3) and Eq. (4), $\gamma\left(\text{Li/LiCl}\right)$ represents the interface formation energy of a specific Li/LiCl interface configuration, while $E\left(\text{Li/LiCl}\right)$ denotes the calculated energy associated with that configuration. The bulk energies of substances Li and LiCl are denoted by $E(\text{Li})$ and $E(\text{LiCl})$, respectively, with $n_{\text{Li}}$ and $n_{\text{LiCl}}$ representing the numbers of Li and LiCl within this Li/LiCl interface configuration. Under the condition of a fixed Li/LiCl interface configuration, altering $n_{\text{Li}}$ leads to varying values of $\gamma\left(\text{Li/LiCl}\right)$ due to the presence of strain energy $\sigma$, which establishes a linear relationship between $\gamma\left(\text{Li/LiCl}\right)$ and $n_{\text{Li}}$. The slope of linear curve represents the interface energy $\gamma^{\text{True}}\left(\text{Li/LiCl}\right)$.

The slab of $Li_9Al_4$(400) surface was constructed for the calculation of Li adsorption energy. The derivation of Li adsorption energy $E_{\text{ad}}$ was listed as follows.

$$E_{\text{ad}} = E_{\text{total}} - \left(E_{\text{slab}} + E_{\text{Li−atom}}\right) \quad (5)$$

In Eq. (5), $E_{\text{total}}$ denotes the energy of the $Li_9Al_4$(400) surface with one adsorbed Li atom, while $E_{\text{slab}}$ representing the energy of $Li_9Al_4$(400) surface and $E_{\text{Li−atom}}$ representing the energy of a free Li atom.

Molecular dynamics simulations of Li/LPSC and $Li_9Al_4$/LPSC interfaces were carried out using ab initio molecular dynamics (AIMD) with a Γ-point-only grid and a cutoff energy of 280 eV. The models of the Li/LPSC and $Li_9Al_4$/LPSC interfaces were constructed by aligning the (100) surface of LPSC with the (100) surface of Li and the (400) surface of $Li_9Al_4$, respectively. The simulations were conducted in the NVT ensemble with a time step of 2 fs over a period of 50 ps at 300 K. RDF profiles of the P-S pair were then derived using the code implemented in the vasppy package.

## Reporting summary

Further information on research design is available in the Nature Portfolio Reporting Summary linked to this article.

## Data availability

The datasets generated during and/or analyzed during the current study are available from the corresponding author on request.

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

## Acknowledgements

Prof. J. P. Tu acknowledges the funding support from the National Natural Science Foundation of China (grant. No. U20A20126, 51971201) and the Key Research and Development Program of Zhejiang Province (2022C01071). Dr. Y. Zhong acknowledges the funding support from the National Natural Science Foundation of China (grant. No. 52103350). The authors thank Dr. Yangjian Lin from the Instrumentation and Service Center for Physical Sciences and Prof. Yuxi Xu from the School of Engineering at Westlake University for their assistance in cryo-FIB-SEM and cryo-TEM measurements.

## Author contributions

Y.Z. and J.P.T. conceived the project and designed the experiment. J.R.L. and H.S. carried out the experiments. H.S. performed the theoretical study with the help of Y.L., X.H.G, J.E.K., M.K.W., C.X.L., and X.L.W. participated in the discussion of the data. H.S., J.R.L., and Y.Z. wrote the initial manuscript with the input of all authors. Y.Z. and J.P.T revised the manuscript and directed the work.

## Competing interests

The authors declare no competing interests.
