## [Peer Review File · Nature Communications]

A scalable Li-Al-Cl stratified structure for stable all-solid-state lithium metal batteriesREVIEWER COMMENTS

Reviewer #1 (Remarks to the Author):

This manuscript presents an innovative negative electrode/SEI integrated material featuring a Li-Al-Cl stratified structure, aimed at enhancing the stability of the Li/sulfide electrolyte interface. The synthesis procedures employed for this material are amenable to easy scalability, thereby demonstrating significant potential for industrial applications. Additionally, the intriguing phase separation observed during the rolling process of the Li/nano-AlCl₃ complex adds an interesting dimension to the study. This phenomenon holds the potential to inspire further research endeavors in the realms of nanotechnology and energy storage. Before the publication of this manuscript, it is necessary to address the following concerns.

1. Can this stratified structure still retain its specific elemental distribution during the electrochemical alloying process (lithiation/delithiation)?
2. It is comprehensible that the Li₉Al₄ phase, with its monoclinic crystal structure, tends to form an incoherent interface with lithium. However, to substantiate the claim that the high interfacial energy of an incoherent interface serves as the thermodynamic driving force for phase separation, the authors should provide comparative results from other lithium alloys featuring a similar crystal type with Li metal. For example, considering the face-centered-cubic (FCC) crystal structure of the Li₂₂Sn₅ phase, according to the authors' theory, the melting complex of Li₂₂Sn₅/Li should not experience significant phase separation. Including such comparative data would strengthen the argument and provide a more comprehensive understanding of the underlying mechanisms.
3. The nano-size effect has been well investigated in the area of alloy preparation. Apart from the high interfacial energy provided by the high specific area of nano particles, the increasing numbers of exposed surfaces in the nano particles can enhance the atom diffusion. The authors should add this point to the discussion of the formation mechanism.
4. Is there any other Li-richest alloys may separate from melting Li matrix except Li₉Al₄?
5. From high-resolution TEM images in the SI, it is evident that in certain regions, lattice patterns overlap, suggesting that samples prepared using FIB may be relatively thick. Typically, FIB is employed to fabricate TEM samples with thicknesses in the range of several tens of nanometers, but in this paper, the thickness of the samples appears to be larger. Why not continue thinning the samples for observation at higher resolutions?
6. In Fig.S7, why do the XRD peaks of metallic lithium exhibit different relative intensity from those indicated in the standard PDF reference?

Reviewer #2 (Remarks to the Author):

The manuscript titled A Scalable Li-Al-Cl Stratified Structure Driven by the Strain-Activated Phase Separation to Stabilize Li/Sulfide Solid Electrolyte Interface submitted to Nature Communications has examined the possibility of scalable technique for all-solid-state lithium metal batteries. The manuscript can be considered for publication if the reviewer addresses the following comments and makes needful modifications to the manuscript.

1. A few electrochemical properties of the LPSC solid electrolyte should be given, like

conductivity and phase

2. The overpotential for the CCD measurement for bare Li cell and LACSS cell looks almost similar or LACSS has more! How does the author comment on it?
3. EIS of the symmetric cells is a must for identifying the interfacial analysis also a cross-sectional SEM is advisable?
4. Comparing the cross-section SEM image in Fig 1b and c why do we see a thickness increment of Li after rolling?
5. The dissociation temperature of AlCl_3 is near 650°C . Would it be more scalable to disperse AlCl_3 into molten lithium? what special difference are we expecting in the particular method?
6. The winding and rolling still leave bare Li that can be in contact with SSE in the cell. Does it sometimes lead to an increased localized current density and short circuit?
7. Authors should be very careful with typos, especially in units and data: third line in the "Performance of ASSLMs using the LACSS (page numbers should have been provided!) "...and high-voltage NCM83125 positive electrode with a mass loading of 8.9 mg/cm^2 " was constructed and tested at room temperature."

Reviewer #3 (Remarks to the Author):

In this manuscript, the authors synthesized a Li metal anode with Li_9Al_4 and LiCl species embedded that serve as protective interlayer. In general, this is an intriguing work that demonstrated Li-containing species rearrangement during rolling process and delivered good electrochemical performance. But there are some comments that need to be addressed before publication.

Comments:

1. After the formation of interlayer, are Al and Cl species uniformly covering the LACSS surface? Is there any spot where fresh Li metal can be exposed? During cell cycling, is newly grown Li metal being deposited on the interlayer or underneath it?
2. A good SEI in principle needs to be both electronically insulating and ionically conductive. In this work, Li_9Al_4 is electronically conductive while LiCl is ionically insulating. Neither of these components falls within the category of good SEI species. Can the authors comment on whether Li_9Al_4 and LiCl are qualified for good SEI under these criteria?
3. How thick is the "enriched" interlayer? The authors have conducted several characterizations to demonstrate the phase gradients of Li_9Al_4 and LiCl . However, the length scales are quite different in TOFSIMS, XPS depth profiling and cryo-TEM. How did the authors conclude the presence of phase separation by cryo-TEM results when the results were only probing $\sim 200 \text{ nm}$ in depth/thickness (Figure 2e)?
4. During TEM sample preparation, was there protective layer deposited on top on the lamella? There seemed to be no protective layer from HADDF image in Figure 2b. If there is no protective layer on top, how did the authors prevent the top surface of the lamella being milled away during sample preparation? In this case, the "surface" in Figure 2b was not the original surface of LACSS electrode anymore.
5. In Figure 2b, can the authors show the line scan profiles of Al and Cl elements to demonstrate the phase separation of those species?
6. Is there an equation or formula used to generate the plot in Figure 3b? In Figure 3c, the authors hypothesized that defects play an important role during atom diffusion. Have the authors observed any defects from TEM images that could be used as evidence for this

hypothesis?

7. In Figure 4b, the LACSS symmetric cell experienced a current density of 4mA/cm at the endpoints of CV scans without the sign of shorting. However, in Figure 4d, the LACSS symmetric cell showed immediate voltage drop when current density reached 2.5mA/cm². Can the author explain why LACSS cell could withstand 4 mA/cm² during CV scans?

8. In Figure 4c, the symmetric cells using bare Li metal shorted immediately when the current density reached 1 mA/cm², but the symmetric cells using bare Li metal was still able to be cycled for 10h in Figure 4g with the same current density. Where is such discrepancy coming from?

9. For the XPS results on the cycled cells, are Figure S15 and S16 collected on the surface of negative electrodes or the SSE pellets? How did the authors separate the SSE pellets from anodes with structural integrity after cycling? Why would the shoulder peak at low energy in Figure S16a be identified as "Li-Al-S" species when the Al signature is almost absent (Figure 16c)?

10. The authors claimed that "...an extremely high average Young's modulus of 64 GPa, which plays a significant role in depressing potential Li penetration..." However, Li penetration is only harmful when it happens within SSE layer. Li penetration does not have much to do with the anode itself.

11. What is the full cell capacity in mAh/cm²? What is the N/P ratio? The deposition of Li metal for only 0.5 mAh/cm²

12. Why did the full cell need activation process? What was occurring in the cell during the activation cycles? Was there any external pressure applied during all the electrochemical tests?

Some minor ones:

1. The XPS peak fitting in Figure S3 is incorrect. Al peaks and AlCl peaks should not be totally covered under the other species.

2. Equation 1 needs to list Li metal as the reactant and be balanced.

3. The schematic in Figure 3a is a bit confusing as it seems to indicate LACSS is sandwiched between Li₉Al₄ and LiCl phases.

4. The overall language could use some major improvement.

Reviewer #1

This manuscript presents an innovative negative electrode/SEI integrated material featuring a Li-Al-Cl stratified structure, aimed at enhancing the stability of the Li/sulfide electrolyte interface. The synthesis procedures employed for this material are amenable to easy scalability, thereby demonstrating significant potential for industrial applications. Additionally, the intriguing phase separation observed during the rolling process of the Li/nano-AlCl₃ complex adds an interesting dimension to the study. This phenomenon holds the potential to inspire further research endeavors in the realms of nanotechnology and energy storage. Before the publication of this manuscript, it is necessary to address the following concerns.

Reply: Thanks for your valuable comments! We have addressed each point in our responses below, and we believe that your kind suggestions significantly enhance the overall quality of this work.

Comment 1. *Can this stratified structure still retain its specific elemental distribution during the electrochemical alloying process (lithiation/delithiation)?*

Response: Thank you for your thoughtful comments!

As illustrated by cross-sectional SEM and EDS results of cycled LACSS/LPSC interface (**Fig. R1.1**, also referred to as **Fig. 5b** in the manuscript), the Al and Cl elements still enrich on the surface of LACSS and diluted in the bulk. Additionally, we have confirmed that Li metal deposits beneath the surface of the LACSS, as evidenced by the continued presence of Al and Cl elements on the surface of LACSS (**Fig. R1.2**, also referred to as **Fig. S26** in the SI) even after plating with 2 mAh cm⁻² of Li (~10 μm thick). These results suggest that our LACSS can retain its specific elemental distribution during the electrochemical alloying process.

Fig. R1.1 Cross-sectional SEM image of the LACSS/LPSC interface and corresponding EDS mappings of Al, Cl and P.

Fig. R1.2 Cross-sectional SEM image and the corresponding EDS mapping results of the LACSS/LPSC interface after the deposition of 2 mAh cm⁻² Li.

The corresponding revisions in our paper are listed below:

-Manuscript, Results and Discussion, Stabilization of the Li/SSE interface by the LACSS

“...Furthermore, to investigate the influence of LACSS's distinctive structural characteristics on Li plating behavior, we deposited 2 mAh cm⁻² of metallic lithium onto the LACSS electrode at a current density of 0.1 mA cm⁻². Intriguingly, even after such a substantial lithium deposition, the enrichment of Al and Cl persisted on the surface of the LACSS, as evidenced by the EDS mapping results. Additionally, no intrusion of metallic lithium into the LPSC was observed (Fig. S26). These findings suggest that Li deposition occurs beneath the Al/Cl-enriched layer of the LACSS.”

Comment 2. *It is comprehensible that the Li_9Al_4 phase, with its monoclinic crystal structure, tends to form an incoherent interface with lithium. However, to substantiate the claim that the high interfacial energy of an incoherent interface serves as the thermodynamic driving force for phase separation, the authors should provide comparative results from other lithium alloys featuring a similar crystal type with Li metal. For example, considering the face-centered-cubic (FCC) crystal structure of the $Li_{22}Sn_5$ phase, according to the authors' theory, the melting complex of $Li_{22}Sn_5/Li$ should not experience significant phase separation. Including such comparative data would strengthen the argument and provide a more comprehensive understanding of the underlying mechanisms.*

Response: Thank you for your valuable comments!

Following your suggestions, we introduced Sn into the molten Li matrix to examine the phase distribution behavior between $Li_{22}Sn_5$ and the Li matrix. The atomic ratio of added Sn is 1.9 at% to ensure an excess of Li metal. As depicted in **Fig. R1.3a** (also referred to as **Fig. S18a** in the SI), SEM and EDS mapping results indicate that the complex of $Li_{22}Sn_5/Li$ prepared by melting-cooling demonstrate no surface enrichment of Sn element, implying the absence of phase separation. Furthermore, we investigated the phase distribution behavior between Li-In/Li-Mg alloys and the Li matrix. Similarly, the atomic ratio of added In (Mg) metals is 1.9 at% to ensure an excess of Li metal. As shown in **Fig. R1.3b-c** (also referred to as **Fig. S18b-c** in the SI), the Li-In and Li-Mg alloys distribute homogeneously within the Li matrix without phase separation. This observation strengthens our theory, as the Li-richest phase of Li-In alloys is $Li_{13}In_3$ with an FCC crystal structure, and Mg forms a solid solution with Li. Neither of these alloy phases possesses significant interfacial energy with the Li matrix.

Fig. R1.3 Cross-sectional SEM images and corresponding EDS mappings of different Li alloys/Li complexes prepared by melting-cooling. **a** Cross-sectional SEM image and the EDS mapping of Li-Sn/Li complexes prepared by melting-cooling. **b** Cross-sectional SEM image and the EDS mapping of Li-In/Li complexes prepared by melting-cooling. **c** Cross-sectional SEM image and the EDS mapping of Li-Mg/Li complexes prepared by melting-cooling.

The corresponding revisions in our paper are listed below:

-Manuscript, Results and Discussion, Formation Mechanism of the LACSS

“...Due to their lattice mismatch, the coupling of Li and Li_9Al_4 is likely to present an incoherent interface with a large $\tau_{\text{Li-Li}_9\text{Al}_4}$, which serves as a thermodynamic driving force for the phase separation of Li_9Al_4 and Li. To experimentally validate this hypothesis, we separately introduced Al, In, Sn, and Mg into the molten Li metal matrix. The atomic ratio of the incorporated metals (Al, In, Sn, Mg) was controlled to 1.9 at% to ensure an excess of lithium metal. Upon cooling, only Li_9Al_4 , which exhibits incoherent interface with the Li matrix, displays phase separation among the prepared mixtures (Figs. S15-S17). In contrast, the other Li-richest alloys (Li-In, Li-Sn, Li-Mg), possessing a body-centered cubic structure similar to that of Li metal, exhibit no noticeable discrepancy between their surface and bulk distributions (Fig. S18).”

Comment 3. *The nano-size effect has been well investigated in the area of alloy preparation. Apart from the high interfacial energy provided by the high specific area of nano particles, the increasing numbers of exposed surfaces in the nano particles can enhance the atom diffusion. The authors should add this point to the discussion of the formation mechanism.*

Response: Thank you for your valuable comments!

The point of “increasing numbers of exposed surfaces can enhance the atom diffusion” has been added in the discussion of the formation mechanism and highlighted in yellow.

Below is the corresponding revision in our paper.

-Manuscript, Results and Discussion, Formation Mechanism of the LACSS

“...Additionally, in line with previous findings, it has been reported that the increased number of exposed surfaces in small-size particles can augment atom diffusion across phase boundaries.”

Comment 4. *Is there any other Li-richest alloys may separate from melting Li matrix except Li_9Al_4 ?*

Response: Thank you for your thoughtful comments!

In our investigation, we thoroughly reviewed all reported phase diagrams of binary Li alloys and discovered nearly every Li-richest alloy exhibited a cubic structure, except for Li_9Al_4 (monoclinic) and Li_5B_4 (Trigonal). However, upon examining previous reports on $\text{Li}_5\text{B}_4/\text{Li}$ complex,¹ we observed that Li_5B_4 prepared via melting displayed a filamentous structure unevenly dispersed within the Li matrix, unlike the distinct distribution differences between surface and bulk observed in $\text{Li}_9\text{Al}_4/\text{Li}$ complex. Consequently, we propose that there may be no other Li-richest alloys, at least among binary alloys, capable of segregating from the molten Li matrix except Li_9Al_4 in theory.

Comment 5. *From high-resolution TEM images in the SI, it is evident that in certain regions, lattice patterns overlap, suggesting that samples prepared using FIB may be relatively thick. Typically, FIB is employed to fabricate TEM samples with thicknesses in the range of several tens of nanometers, but in this paper, the thickness of the samples appears to be larger. Why not continue thinning the samples for observation at higher resolutions?*

Response: Thank you for your thoughtful comments!

If the thinning process continues, the sample may experience severe perforation or extensive detachment from the Pt protective layer. To preserve the original structure of the sample as much as possible, we chose not to continue thinning the sample.

The FIB instrument equipped in this study utilizes a Ga ion source. At room temperature, metallic lithium undergoes alloying reactions with Ga, and the heat generated during thinning at room temperature can damage the structure of metallic Li-containing samples.² Therefore, the sample preparation was conducted at -195°C. However, even at that low temperature, we discovered that the FIB can cause mechanical damage to the sample, especially in the later stages of thinning, leading to partial detachment of the sample, as demonstrated in **Fig. R1.4** (also referred to as **Fig. S10a** in the SI). If thinning were to continue on this sample, complete detachment of the sample would inevitably ensue, rendering it unavailable for TEM characterization. Moreover, prolonged exposure to the FIB could adversely affect the sample's microstructure. Taking these factors into consideration, we opted to halt the thinning process and utilize cryo-TEM to observe the sample at the point when detachment first commenced.

Fig. R1.4 Front-view SEM image of the lamella prepared via FIB for TEM observation.

Comment 6. *In Fig.S7, why do the XRD peaks of metallic lithium exhibit different relative intensity from those indicated in the standard PDF reference?*

Response: Thank you for your thoughtful comments!

We have provided additional XRD patterns of Li foils subjected to various treatments. As illustrated in **Fig. R1.5**, the untreated Li foil exhibits a preferential orientation of the (200) crystal plane. Following surface polishing to eliminate the oxide layer, the XRD spectrum of the Li foil closely resembles the standard XRD spectrum of Li metal. Upon undergoing the rolling process subsequent to polishing, the XRD pattern of the Li foil once again exhibits a preferential orientation of the (200) crystal plane. This observation suggests that the rolling procedure induces alterations in the orientation of Li grains, leading to the manifestation of a preferential alignment of the (200) crystal plane within the Li foil. A similar phenomenon has been documented in prior studies.³

Fig. R1.5 XRD patterns of Li foils before and after different mechanical treatments.

Reviewer #2

The manuscript titled *A Scalable Li-Al-Cl Stratified Structure Driven by the Strain-Activated Phase Separation to Stabilize Li/Sulfide Solid Electrolyte Interface* submitted to *Nature Communications* has examined the possibility of scalable technique for all-solid-state lithium metal batteries. The manuscript can be considered for publication if the reviewer addresses the following comments and makes needful modifications to the manuscript.

Reply: Thank you for your thoughtful comments and thorough review! We have taken into account each of your points in our responses below, and we are confident that your valuable suggestions have greatly improved the quality of this work.

Comment 1. A few electrochemical properties of the LPSC solid electrolyte should be given, like conductivity and phase.

Response: Thank you for your valuable comments!

We conducted XRD analysis on the synthesized LPSC solid electrolytes. As depicted in **Fig. R2.1a** (also referred to as **Fig. S28a** in the SI), the XRD pattern of the synthesized LPSC solid electrolyte exhibits a peak distribution pattern similar to that reported for $\text{Li}_6\text{PS}_5\text{Cl}$. Additionally, the ionic conductivity of the LPSC was measured to be 7.9 mS cm^{-1} using EIS, as shown in **Fig. R2.1b** (also referred to as **Fig. S28b** in the SI).

Fig. R2.1 Characterizations of the LPSC electrolyte. **a** XRD pattern of the LPSC powder. **b** Nyquist plot of the cold-pressed LPSC electrolyte pellet at $25 \text{ }^\circ\text{C}$.

The corresponding revisions in our paper are presented below:

-Manuscript, Methods, Cell assembly and electrochemical measurements

"...The XRD pattern of the synthesized LPSC is provided in Fig. S28a. The ionic conductivity of the synthesized LPSC was measured to be 7.9 mS cm^{-1} (Fig. S28b)."

Comment 2. *The overpotential for the CCD measurement for bare Li cell and LACSS cell looks almost similar or LACSS has more! How does the author comment on it?*

Response: Thank you for your thoughtful comments!

Firstly, it is understandable that the Li symmetric cell and the LACSS symmetric cell exhibit similar overpotentials. From the EIS measurements conducted immediately after assembling the bare Li symmetric cell and the LACSS symmetric cell (**Fig. R2.2**, also referred to as **Fig. S22** in the SI), it can be observed that the impedance of the LACSS symmetric cell is almost similar with that of the Li symmetric cell. This phenomenon can be elucidated by two primary factors:

(1) Based on the previous report,⁴ the high-Cl content LPSC is relatively stable towards Li metal and the Li/LPSC interface in the freshly assembled symmetric cells has not undergone complete reaction yet.

(2) In comparison to the "fresh" Li/LPSC interface, the incorporation of LACSS introduces an additional SEI layer with a thickness of ~100 nm between the negative electrode and the SSE. Nevertheless, owing to the low thickness of this SEI and its commendably rapid ion transport kinetics, it does not engender a noteworthy escalation in the overall impedance.

After 10 cycles of discharging/charging at 0.5 mA cm^{-2} and 0.5 mAh cm^{-2} , the LACSS symmetric cell showcases a smaller increase of the resistance than the bare Li symmetric cell, indicating the SEI of the LACSS can decrease the side reactions (**Fig. R2.2**). The role of the LACSS in decreasing side reactions can also be inferred from the voltage profiles of bare Li symmetric cell and LACSS symmetric cell during cycling at moderate current density (0.5 mA cm^{-2} , 0.5 mAh cm^{-2}). Initially, both bare Li symmetric cell and LACSS symmetric cell exhibit

similar overpotentials at the onset of cycling. As the cycling progresses, the increase in impedance of the bare Li symmetric cell exceeds that of the LACSS symmetric cell (**Fig. R2.3**, also referred to as **Fig. 4e-f** in the manuscript).

Secondly, by comparing the voltage profiles of bare Li symmetric cell and LACSS symmetric cell during CCD tests (**Fig. R2.4**), we observed that prior to the current density reaching 0.8 mA cm^{-2} , the overpotential of the LACSS symmetric cell is slightly lower than that of the bare Li symmetric cell. Interestingly, as the current density reaches 0.8 mA cm^{-2} (approaching the CCD of the bare Li symmetric cell), the overpotential of the bare Li symmetric cell begins to become lower than that of the LACSS symmetric cell. This phenomenon may be attributed to the infiltration of some lithium into the SSE in the bare Li symmetric cell (without causing a direct short circuit), resulting in a shortened effective distance between electrodes (reduced Ohmic impedance).

Fig. R2.2 EIS spectra of the bare Li symmetric cell and the LACSS symmetric cell before and after cycling at $0.5 \text{ mA cm}^{-2}/0.5 \text{ mAh cm}^{-2}$ for 10 cycles.

Fig. R2.3 Galvanostatic cycling performance of the bare Li symmetric cell and the LACSS symmetric cell at $0.5 \text{ mA cm}^{-2}/0.5 \text{ mAh cm}^{-2}$.

Fig. R2.4 Comparison of the voltage profiles of bare Li symmetric cell and LACSS symmetric cell during CCD tests.

Comment 3. *EIS of the symmetric cells is a must for identifying the interfacial analysis also a cross-sectional SEM is advisable?*

Response: Thank you for your valuable comments!

We have provided the EIS spectra of the bare Li symmetric cell and the LACSS symmetric cell before and after cycling at $0.5 \text{ mA cm}^{-2}/0.5 \text{ mAh cm}^{-2}$ for 10 cycles. As displayed in **Fig. R2.5** (also referred to as **Fig. S22** in the SI), the bare Li symmetric cell and the LACSS symmetric cell demonstrates an almost similar resistance before cycling. After 10 cycles of discharging/charging, the LACSS symmetric cell showcases a smaller increase of the resistance than the Li symmetric cell, indicating the SEI of the LACSS can decrease the side reactions. Additionally, as the cross-sectional SEM images of negative electrode/SSE interface after cycling at $0.5 \text{ mA cm}^{-2}/0.5 \text{ mAh cm}^{-2}$ for 10 cycles illustrate, the in-situ formed SEI in the Li symmetric cell became incapable of impeding lithium penetration into the SSE, resulting in severe structural damage to SSEs. In sharp contrast, with the intervention of the LACSS, mechanical failure of the SSE was effectively mitigated, as revealed by the structural integrity of the LPSC (**Fig. R2.6a-b**, also referred to as **Fig. 5a-b** in the manuscript).

Fig. R2.5 EIS spectra of the bare Li symmetric cell and the LACSS symmetric cell before and after cycling at $0.5 \text{ mA cm}^{-2}/0.5 \text{ mAh cm}^{-2}$ for 10 cycles.

Fig. R2.6 Cross-sectional SEM image of the electrode/SSE interface after cycling. **a** Cross-sectional SEM image of the bare Li/LPSC interface and corresponding EDS mapping results of Cl and P. **b** Cross-sectional SEM image of the LACSS/LPSC interface and corresponding EDS mapping results of Al, Cl and P.

The corresponding revisions in our paper are presented below:

-Manuscript, Results and Discussion, Stabilization of the Li/SSE Interface by the LACSS

“Post-mortem analyses were then performed on the bare Li/LPSC and LACSS/LPSC interfaces. As illustrated in Fig. S22, the electrochemical impedance spectra (EIS) of the bare Li symmetric cell and the LACSS symmetric cell demonstrated almost similar resistances before cycling. After 10 cycles of charging/discharging at $0.5 \text{ mA cm}^{-2}/0.5 \text{ mAh cm}^{-2}$, the LACSS symmetric cell showed a smaller increase in the resistance than the Li symmetric cell, indicating that the SEI of the LACSS can decrease the side reactions.

...As shown in the cross-sectional SEM images of the bare Li/LPSC and LACSS/LPSC

interfaces, after 10 cycles of Li stripping/plating, the in-situ formed SEI in the bare Li symmetric cell became incapable of impeding lithium penetration into the LPSC, resulting in severe structural damage (Fig. 5a). In sharp contrast, with the intervention of the LACSS, mechanical failure induced of the SSE was effectively mitigated, as revealed by the structural integrity of the LPSC. Also, as showcased by the EDS mapping results, Al and Cl are still enriched on the surface of the LACSS and diluted in bulk (Fig. 5b), which indicates that the LACSS maintains its special structure during the electrochemical process.”

Comment 4. Comparing the cross-section SEM image in Fig 1b and c why do we see a thickness increment of Li after rolling?

Response: Thank you for your thoughtful comments!

Fig. 1b (denoted as **Fig. R2.7b**) is the SEM image of **AlCl₃-deposited Li**. Then, the foil of AlCl₃-deposited Li will undergo a winding process, resulting in a **roll-like composite with a very large thickness**, as illustrated in **Fig. R2.8** (also referred to as **Fig. S4** in the SI). The final LACSS (depicted in Fig. 1c-d, also denoted as Fig. R2.7c-d in this response letter) is prepared via the rolling of the AlCl₃-deposited Li metal after winding rather than directly rolling the AlCl₃-deposited Li metal. Our rolling process effectively reduces the thickness of the foil compared to that of the AlCl₃-deposited Li metal after winding. Moreover, by applying increased pressure during the rolling process, we can produce LACSS sample with a thinner thickness. The SEM images and corresponding EDS results of the LACSS sample with a thinner thickness than that presented in **Fig. 1c** are elucidated in **Fig. R2.9** (also referred to as **Fig. S6** in the SI).

Fig. R2.7 **a** Top-view SEM image of bare Li foil. **b** Top and cross-sectional views of the AlCl_3 -deposited Li foil. **c** Top and cross-sectional views of the prepared LACSS, and **d** the correspond EDS mapping results of Al and Cl elements. **e** Schematic diagram of the LACSS preparation process.

Fig. R2.8 Cross-sectional SEM image of the AlCl_3 -deposited Li metal after winding.

Fig. R2.9 Cross-sectional SEM image and the corresponding EDS mapping results of the LACSS with a thickness of $\sim 100 \mu\text{m}$.

Comment 5. *The dissociation temperature of AlCl_3 is near 650°C . Would it be more scalable to disperse AlCl_3 into molten lithium? what special difference are we expecting in the particular method?*

Response: Thank you for your thoughtful comments!

While mixing the reactants with molten Li represents a viable strategy for large-scale production of negative electrode materials, our mechanical-chemical approach for preparing LACSS offers several notable advantages. Firstly, the direct melting process entails heating and stirring metallic Li and AlCl_3 at elevated temperatures over an extended duration to ensure a complete reaction, presenting operational inconveniences. Secondly, due to the higher melting point of metallic Li compared to the sublimating point of AlCl_3 , the addition of AlCl_3 to the molten lithium matrix may result in direct sublimation of some AlCl_3 , posing challenges in controlling the AlCl_3 dosage. Most importantly, during our revision, we performed an experiment involving the addition of AlCl_3 into the molten Li matrix. During the reaction process between AlCl_3 and molten Li, a dark grey layer will form on the surface of the molten Li due to the phase separation between the reaction products ($\text{Li}_9\text{Al}_4/\text{LiCl}$) and Li matrix. Additionally, some larger particles of reaction products are unevenly embedded on the surface of Li matrix or directly detached from the molten Li (**Fig. R2.10a**). This uneven surface distribution caused by the uneven particle size of the reaction products poses challenges for controlled preparation.

What's more, this AlCl_3/Li complex will demonstrate an ellipsoidal shape after cooling, which requires further mechanical process to be applied as an electrode foil. However, the mechanical processing, such as rolling, would destroy the surface structure of this complex by exposing a significant amount of metallic Li (Fig. R2.10b), which may be detrimental to the electrochemical performance. Besides, it can be observed that there are a lot of large black particles unevenly distributing on the surface of this rolled complex. In contrast, the unique SEI/anode integrated structure of the LACSS electrode is in-situ formed during the rolling, which demonstrates a potential for controllable and scalable production.

Fig. R2.10 **a** AlCl_3/Li complex prepared via dispersing AlCl_3 into molten Li. **b** A foil of AlCl_3/Li complex prepared by further rolling.

Comment 6. *The winding and rolling still leave bare Li that can be in contact with SSE in the cell. Does it sometimes lead to an increased localized current density and short circuit?*

Response: Thank you for your thoughtful comments!

We have evaluated phase distributions from the surface region to the bulk region in the LACSS stepwise via selecting regions 1-3 as highlighted by the squares in **Fig. R2.11** (also referred to as **Fig. 2e-h** in the manuscript).

Fig. R2.11 HRTEM image of the LACSS where three regions (regions 1–3) are highlighted by blue squares and FFT images from region 1 to region 3.

Based on the FFT results from regions 1 and 2, it is discernible that within the Al and Cl enriched layer nearest to the outer surface of LACSS, there is an absence of corresponding diffraction spots characteristic of metallic Li. Hence, we are inclined to believe that there is no exposed metallic lithium present on the outer surface of the LACSS to be in contact with SSE. Furthermore, even if there were minute traces of residual lithium on the surface, these remnants would swiftly react with the SSE during the battery assembly process. Consequently, they would have minimal impact on the electrochemical performance.

To substantiate the aforementioned assertions, we conducted multiple CCD tests on symmetric cells utilizing LACSS to provide additional electrochemical evidence, as illustrated in **Fig. R2.12**. It is noteworthy that all CCD values acquired from the symmetric cell employing LACSS exceed 2.0 mA cm^{-2} , highlighting the robust reproducibility of the electrochemical performance demonstrated by LACSS.

Fig. R2.12 Voltage profiles of other two batches of LACSS symmetric cells during galvanostatic discharging/charging tests at step-increased current densities. The CCD values are determined to be 2.3 mA cm^{-2} for the batch 2 and 2.2 mA cm^{-2} for the batch 3.

Comment 7. *Authors should be very careful with typos, especially in units and data: third line in the "Performance of ASSLMBs using the LACSS (page numbers should have been provided!)..and high-voltage NCM83125 positive electrode with a mass loading of 8.9 "mg/cm⁻²" was constructed and tested at room temperature."*

Response: Thank you for your careful review!

We have checked the whole manuscript to avoid the appearance with typos, especially those in units and data (All the revisions on typos are remarked in yellow in the manuscript). In addition, we supplied the page number for the convenience of reading.

Reviewer #3

In this manuscript, the authors synthesized a Li metal anode with Li_9Al_4 and LiCl species embedded that serve as protective interlayer. In general, this is an intriguing work that demonstrated Li-containing species rearrangement during rolling process and delivered good electrochemical performance. But there are some comments that need to be addressed before publication.

Reply: Thank you for your valuable feedback and thorough review! We have taken into account each of your points in our responses below, and we believe that your thoughtful suggestions have greatly improved the quality of this work.

Comments

Comment 1. *After the formation of interlayer, are Al and Cl species uniformly covering the LACSS surface? Is there any spot where fresh Li metal can be exposed? During cell cycling, is newly grown Li metal being deposited on the interlayer or underneath it?*

Response: Thank you for your critical comments!

Based on the signals of $\text{Li}_5\text{Al}_2\text{O}^+$ and Li_2Cl^+ ions in TOF-SIMS (**Fig. R3.1**, also referred to as **Fig. 1g** in the manuscript), as well as the EDS mapping results obtained from TEM (**Fig. R3.2a-b**, also referred to as **Fig. 2b-c** in the manuscript), we posit that Al and Cl are uniformly distributed on the surface of the LACSS. Additionally, we have investigated phase distributions from the surface region to the bulk region in the LACSS stepwise via selecting regions 1-3 as highlighted by the squares in **Fig. R3.2d-g** (also referred to as **Fig. 2e-h** in the manuscript). According to the FFT images of regions 1 and 2 in **Fig. R3.2e-f**, it is observed that within the Al and Cl enriched layer, closest to the outer surface, there is no corresponding diffraction spot indicative of metallic lithium. Therefore, we are inclined to believe that there is no exposed metallic lithium on the outer surface of the LACSS.

Fig. R3.1 Depth-profiling TOF-SIMS of $\text{Li}_5\text{Al}_2\text{O}^+$, Li_2Cl^+ , and Li_3^+ secondary ion fragments on the surface of the LACSS.

Fig. R3.2 **a** HAADF-TEM image of the LACSS and the corresponding EDS mapping results of Al and Cl. **b** Line scan profiles of Al and Cl elements. **d** SAED result including the bulk and surface regions in LACSS. **d** HRTEM image of the LACSS where three regions (regions 1–3) are highlighted by blue squares. **e-g** FFT images from region 1 to region 3.

Then, to elucidate whether lithium deposition occurs on the interlayer consisted of only $\text{Li}_9\text{Al}_4/\text{LiCl}$ (the SEI of the LACSS) or beneath it, we deposited 2 mAh cm^{-2} of metallic lithium ($\sim 10 \mu\text{m}$ thick) on the LACSS electrode at a current density of 0.1 mA cm^{-2} . As depicted in **Fig. R3.3** (also referred to as **Fig. S26** in the SI), even with such a high-capacity Li deposition, the Al and Cl elements continue to enrich on the surface of the LACSS. Also, the LACSS electrode maintains good contact with the LPSC electrolyte without any observed intrusion of metallic Li into the LPSC. These results suggest that Li deposition occurs beneath the SEI of LACSS. Furtherly, we conducted multiple CCD tests on symmetric cells using the LACSS, as shown in

Fig. R3.4. The results of the CCD tests demonstrate highly reproducible electrochemical performances of the LACSS. This observation implies that there is no exposed metallic Li on the outermost layer of the LACSS, and the electronically insulating LiCl reduces the electronic conductivity of LACSS's SEI, ensuring the deposition of metallic Li beneath the SEI. Otherwise, if metallic Li were to directly deposit on the SEI of the LACSS, its electrochemical performance would not exhibit such a significant enhancement compared to the symmetric cell using bare Li electrode.

Fig. R3.3 Cross-sectional SEM image and the corresponding EDS mapping results of the LACSS/LPSC interface after the deposition of 2 mAh cm^{-2} Li.

Fig. R3.4 Voltage profiles of other two batches of LACSS symmetric cells during galvanostatic discharging/charging tests at step-increased current densities. The CCD values are determined to be 2.3 mA cm^{-2} for the batch 2 and 2.2 mA cm^{-2} for the batch 3.

The corresponding revisions in our paper are listed as follows:

-Manuscript, Results and Discussion, Stabilization of the Li/SSE Interface by the LACSS

“...Furthermore, to investigate the influence of LACSS's distinctive structural characteristics on Li plating behavior, we deposited 2 mAh cm^{-2} of metallic lithium onto the LACSS electrode at a current density of 0.1 mA cm^{-2} . Intriguingly, even after such a substantial lithium deposition,

the enrichment of Al and Cl persisted on the surface of the LACSS, as evidenced by the EDS mapping results (Fig. S26). Additionally, no intrusion of metallic lithium into the LPSC was observed (Fig. S26). These findings suggest that Li deposition occurs beneath the Al/Cl-enriched layer of the LACSS."

Comment 2. *A good SEI in principle needs to be both electronically insulating and ionically conductive. In this work, Li_9Al_4 is electronically conductive while LiCl is ionically insulating. Neither of these components falls within the category of good SEI species. Can the authors comment on whether Li_9Al_4 and LiCl are qualified for good SEI under these criteria?*

Response: Thank you for your critical comments!

If the SEI components (Li_9Al_4 and LiCl) are considered in isolation, Li_9Al_4 , with its high electronic conductivity, and LiCl , with its low ionic conductivity, indeed do not meet the criteria for a desirable SEI as previously suggested by early research. However, when these two components are combined, the high Li diffusivity of Li_9Al_4 compensates for the low ionic conductivity of LiCl . In addition, based on the previous report, The phase interface generated by the two-phase mixture can also serve as a rapid pathway for ion transport.⁵ In the SEI of LACSS, there are a substantial amount of $\text{Li}_9\text{Al}_4/\text{LiCl}$ phase interface, which may enhance the Li^+ transport kinetics. Simultaneously, the insulating nature of LiCl reduces the overall electronic conductivity of the SEI. These, in turn, ensure that the interface maintains high Li^+ flux and low electron flux.

Furthermore, the SEI composed of Li alloys and LiCl has been documented by the research group of Prof. Linda Nazar.⁶ According to their findings, the presence of Li alloys in the SEI promotes Li^+ transport, while the coincidental formation of insulating LiCl alongside the alloys results in a resistive composite film. They observed that Li metal deposition occurs beneath the SEI containing Li alloys and LiCl . Similarly, as addressed in our response to **Comment 1**, our SEI comprising Li_9Al_4 and LiCl also facilitates Li metal plating underneath the surface of LACSS, which confirms that the LiCl in the SEI can lower the overall electronic conductivity. Moreover, as depicted in **Fig. R3.5** (also referred to as **Fig. S23** in the SI), the

exchange current density of the LACSS symmetric cell (1.11 mA cm^{-2}) after 10 cycles at 0.5 mA cm^{-2} and 0.5 mAh cm^{-2} nearly doubles that of the bare Li symmetric cell (0.69 mA cm^{-2}). This suggests enhanced Li diffusivity through the SEI of the LACSS, indicating satisfactory Li^+ flux and cycling stability of the SEI.

Fig. R3.5 Tafel plots of the symmetric Li|LPSC|Li and LACSS|LPSC|LACSS cell after 10 cycles at $0.5 \text{ mA cm}^{-2}/0.5 \text{ mAh cm}^{-2}$ obtained by linear sweep voltammetry (LSV).

Additionally, with the continued investigation into the Li/SSE interface, more and more researchers have proposed that incorporating lithiophilic components into the SEI can ensure its strong affinity with Li metal.^{7, 8} In this context, the presence of Li_9Al_4 in the SEI, with its high adsorption energy (-1.97 eV) towards Li metal, fulfills this requirement (**Fig. R3.6**, also referred to as **Fig. S27** in the SI).

Fig. R3.6 Adsorption of Li atom on the $\text{Li}_9\text{Al}_4 (400)$ surface. The green balls represent Li atoms in the Li_9Al_4 , the red ball represents the Li atom adsorbed on the Li_9Al_4 surface and the blue balls represent Al atoms. The adsorption energy is calculated to be -1.97 eV .

Comment 3. *How thick is the “enriched” interlayer? The authors have conducted several characterizations to demonstrate the phase gradients of Li_9Al_4 and LiCl . However, the length scales are quite different in TOF-SIMS, XPS depth profiling and cryo-TEM. How did the authors conclude the presence of phase separation by cryo-TEM results when the results were only probing ~200 nm in depth/thickness (Figure 2e)?*

Response: Thanks for your thoughtful comments!

Firstly, we would like to elucidate the logical deduction of phase separation phenomenon within the LACSS. Based on the SEM and EDS results depicted in **Fig. R3.7d** (also referred to as **Fig.1d** in the manuscript), it is evident that there is a significant enhancement of Al and Cl elemental signals (highlighted by the white dashed box) on the surface of the LACSS, extending to approximately nanometer-scale dimensions. However, the low resolution of SEM fails to provide an accurate depiction of the aggregation range of Al/Cl elements. Given the premise established by SEM findings indicating differential elemental aggregation between the bulk and surface of the LACSS, we then conducted cryo-TEM characterization of the LACSS. Results from EDS surface mapping and line scans of the LACSS revealed a conspicuous aggregation of Al and Cl elements extending ~300 nm at the surface of LACSS, corroborating the SEM observations (**Fig. R3.8a-b**, also referred to as **Fig. 2b-c** in the manuscript). Furthermore, FFT was performed on the Al/Cl elemental aggregation layer from outer to inner regions to elucidate the phase distribution behavior. It is observed that within the outermost region of this aggregation layer (~100 nm thick), only diffraction spots corresponding to Li_9Al_4 and LiCl phases are present, with no diffraction spots indicative of metallic lithium (**Fig. R3.8c-g**, also referred to as **Fig. 2d-h** in the manuscript). Based on these findings, we infer the presence of differential phase distribution between the bulk and surface of the LACSS. However, it is noteworthy that there are discernible differences in the trends of phase distribution and elemental distribution within LACSS. While the region of Al/Cl elemental enrichment on the surface of LACSS spans ~ 300 nm in thickness, only the outermost 100 nm thick layer of this region lacks metallic lithium phase, featuring exclusively Li_9Al_4 and LiCl

phases. Hence, we propose that the elemental Al/Cl-enriched interlayer is ~300 nm in thickness, while the layer exclusively comprising $\text{Li}_9\text{Al}_4/\text{LiCl}$ phases is estimated to be ~ 100 nm thick.

Fig. R3.7 **a** Top-view SEM image of bare Li foil. **b** Top and cross-sectional views of the AlCl_3 -deposited Li foil. **c** Top and cross-sectional views of the prepared LACSS, and **d** the correspond EDS mapping results of Al and Cl. **e** Schematic diagram of the LACSS preparation process.

Fig. R3.8 Phase distribution in the LACSS. **a** HAADF-TEM image of the LACSS and the corresponding EDS mapping results of Al and Cl. **b** Line scan profiles of Al and Cl. **d** SAED result including the bulk and surface regions in LACSS. **d** HRTEM image of the LACSS where three regions (regions 1–3) are highlighted by blue squares. **e-g** FFT images from region 1 to region 3.

Secondly, although our study utilized XPS, TOF-SIMS, and cryo-TEM to characterize the enrichment of Al/Cl elements at the surface of the LACSS, we posit that XPS and TOF-SIMS may not provide an accurate length scale for the following reasons: (1). The etching by XPS and TOF-SIMS is inherently destructive, leading to a significant disparity between the information obtained and the actual scenario. (2) While the total etching time during XPS and TOF-SIMS characterization processes is known, the etching rate remains unknown. When samples are etched using ion beams, the etching rate for different samples is not a fixed value but rather varies. For instance, for materials of the same thickness, such as polymers and metals, using the same etching parameters may result in different etching rates due to differences in material properties. Considering this, we propose that only the cryo-TEM characterization in our study can give an accurate length scale of the "enriched" interlayer. Additionally, the primary objective of utilizing TOF-SIMS and XPS characterizations is to mitigate potential fortuitous interpretations solely based on EDS characterization by demonstrating surface enrichment of reaction products Li_9Al_4 and LiCl through multiple forms of characterization techniques.

The corresponding revisions in our paper are listed as follows:

-Manuscript, Results and Discussion, Preparation and Characterization of the LACSS

"...According to the line scan profiles of Al and Cl (Fig. 2c), the thickness of the Al/Cl-enriched layer is about 300 nm. ... These results illustrate that the outermost surface layer of the LACSS, with a thickness of ~100 nm, consists solely of the Li_9Al_4 and LiCl phases. Meanwhile, the unreacted Li metal is concentrated in the bulk with diluted Li_9Al_4 and LiCl phases."

Comment 4. *During TEM sample preparation, was there protective layer deposited on top on the lamella? There seemed to be no protective layer from HADDF image in Figure 2b. If there is no protective layer on top, how did the authors prevent the top surface of the lamella being milled away during sample preparation? In this case, the “surface” in Figure 2b was not the original surface of LACSS electrode anymore.*

Response: Thanks for your thorough review!

We did have a Pt protective layer deposited on the top of the lamella during the TEM sample preparation. As the SEM images of key steps in the TEM sample preparation process illustrate (**Fig. R3.9**, also referred to as **Fig. S9** in the SI), a layer of Pt metal was firstly deposited on the surface of the LACSS sample to protect the sample surface at 25 °C. Subsequently, the sample stage was cooled to -195 °C. Then, the FIB was applied to etch the sample around the protective layer to prepare a lamella of the sample. After that, the lamella was removed using a manipulator arm and transferred onto a Cu grid. Finally, both two sides of the lamella underwent thinning using the FIB to facilitate subsequent TEM observations.

Fig. R3.9 Preparation process of the lamella for TEM characterization via FIB. **a** SEM image of the deposited Pt layer on the surface of LACSS at 25 °C. **b** Top-view SEM image of the etched LACSS around the deposited Pt layer via focused Ga ion beam at -195 °C. **c**, **d** Top-view (**c**) and front-view (**d**) SEM images of the lamella lifted out and welded to the Cu grid via a manipulator arm at -195 °C.

However, during the thinning process, the lamella may not always maintain a tight connection with the Pt protective layer.⁹ Especially in the later stages of thinning, partial detachment of the LACSS sample is prone to occur, as illustrated in **Fig. R3.10a** (also referred to as **Fig. S10a** in the SI). Given the necessity for clear observation of lattice fringes, thinning the sample to the utmost extent is paramount. Consequently, we opted to examine the region where detachment had just commenced, ensuring minimal sample thickness and maximum preservation of the initial chemical information. We apologize for the oversight in not capturing images of the final samples after thinning during the previous TEM characterization, which resulted in the absence of any information related to the Pt protective layer. **To compensate this**, during the revision process, we employed EDS mapping to characterize another LACSS lamella that was relatively thick but had not yet detached from the Pt layer (**Fig. R3.10b**, also referred to as **Fig. S10b** in the SI). As demonstrated in **Fig. R3.11** (also referred to as **Fig. S11** in the SI), Al and Cl elements are still enriched within the depth range of ~300 nm on the surface of the LACSS. This outcome is in good agreement with our previous characterization results in Fig. 2b (denoted as **Fig. R3.8a** in the response letter). Therefore, we assert that the “surface” in **Fig. 2b** can reflect the original surface of LACSS electrode.

Fig. R3.10 Front-view SEM images of the lamellas for TEM observation. **a** SEM image of the lamella detaching from the Pt layer after thinning. **b** SEM image of a relatively thick lamella maintaining a tight connection with Pt layer after thinning.

Fig. R3.11 Characterizations of the lamella displayed in **Fig. R3.10b** via TEM. **a** TEM image of the lamella. **b** Magnified HAADF-TEM image of the lamella and corresponding EDS mapping results of Pt, Al and Cl.

The corresponding revisions in our paper are listed as follows:

-Manuscript, Results and Discussion, Preparation and Characterization of the LACSS

“...The SEM images of key steps during sample preparation are provided in Fig. S9. It should be noted that once the LACSS sample was thinned to a certain extent, its detachment from the Pt layer might occur (Fig. S10a). Given the necessity for clear observation of lattice fringes, thinning the sample to the utmost extent is paramount. Consequently, we opted to examine the region where detachment had just commenced, ensuring minimal sample thickness and maximum preservation of the initial chemical information. As illustrated by the high-angle annular dark-field-transmission electron microscopy (HAADF-TEM) image in Fig. 2b, compared to the bulk of the LACSS lamella, the surface of the LACSS lamella demonstrates enrichment of heavy atoms as indicated by the increased intensity of imaging spots. These heavy atoms are confirmed to be Al and Cl based on the EDS areal mappings. In addition, we employed EDS mapping to characterize another LACSS lamella that was relatively thick but had not yet detached from the Pt layer (Fig. S10b). As shown in Fig. S11, this lamella exhibited an elemental distribution consistent with the region just beginning to detach from the Pt layer, thus affirming the intact preservation of chemical information in our selected region for further TEM characterization.”

Comment 5. In Figure 2b, can the authors show the line scan profiles of Al and Cl elements to demonstrate the phase separation of those species?

Response: Thank you for your valuable comments!

The line scan profiles of Al and Cl elements have been provided in **Fig. R3.12a-b** (also referred to as **Fig. 2b** and **2c** in the manuscript). As demonstrated by the line scan results, the Al and Cl elements display a significant enrichment on the surface of the LACSS, indicating the phase separation of those species.

Fig. R3.12 Line scan profiles of Al and Cl elements. **a** HAADF-TEM image of the LACSS lamella. **b** Line scan profiles of the Al and Cl elements in the area marked by the yellow box in **a**.

Below is the corresponding revision in our manuscript:

-Manuscript, Results and Discussion, Preparation and Characterization of the LACSS

"...According to the line scan profiles of Al and Cl elements (Fig. 2c), the thickness of the Al/Cl-enriched layer is about 300 nm."

Comment 6. *Is there an equation or formula used to generate the plot in Figure 3b? In Figure 3c, the authors hypothesized that defects play an important role during atom diffusion. Have the authors observed any defects from TEM images that could be used as evidence for this hypothesis?*

Response: Thank you for your valuable comments!

Firstly, we would like to answer the question whether there is a formula to generate the plot in **Fig. 3b**. The total interfacial free energy (E_{int}) is the product of unit interfacial energy (τ) and the total area $S_{TOT}(r)$ of the embedded particle (with a radius of r) in the matrix (**Equation R1**):

$$E_{int} = S_{TOT}(r) \cdot \tau \quad \text{Equation R1}$$

Assuming all embedded particles are spherical and have a constant mass of M when added to the substrate, the relationship between particle radius (r) and the number of particles (n) is as follows (**Equation R2**):

$$n = \frac{M}{\frac{4}{3}\rho\pi r^3} \quad \text{Equation R2}$$

The total surface area $S_{TOT}(r)$ is $4n\pi r^2$. Based on Equation R1 and R2, the following relationship between E_{int} and the radius of the embedded particle can be derived (**Equation R3**).

$$E_{int} = \tau \cdot \frac{3M}{\rho r} \quad \text{Equation R3}$$

Based on **Equation R3**, we generate a “ $y = c/x$ ”-shape (c is a constant) plot in **Fig. 3b**. Additionally, since this deduction is based on the premise that all particles are spherical and does not take into account the case of inconsistent interfacial energies at different interfaces, we prefer not to put the derivation of this plot in the main text. Nevertheless, this plot can still represent the relationship between the size of a particle and its total interfacial Gibbs free energy in the most ideal case. Also, the x-axis in **Fig. 3b** should represent the size of the particle. Therefore, we have made the corresponding revision in **Fig. 3b**, as shown in **Fig R3.13**.

Fig. R3.13 Size effect on phase separation in the solid-state matrix.

Secondly, we would like to answer your concerns on defects. Commonly, in order to observe defects, achieving atomic-level resolution stands as a fundamental requirement. This frequently mandates the utilization of higher-resolution STEM alongside extremely thin sample preparations.^{10, 11} Furthermore, it demands that the sample under investigation exhibits stability under the intense irradiation of high-energy electron beams. As mentioned in our response to your **comment 4**, preparing a thin sample capable of defect observation is exceedingly challenging due to the propensity for the detachment between metallic Li and the Pt protective layer during etching. Furthermore, even with cryo-protection, we have reservations regarding the stability of metallic Li under such intense electron beam irradiation in high-resolution STEM. To the best of our knowledge, successful observations of dislocations and point defects within metallic Li have not been reported thus far. Additionally, to track defects induced by strain, the capability to apply pressure in situ within the STEM is indispensable. Also, throughout the whole procedures, maintaining a consistently low temperature of 98 K is crucial to minimize damage inflicted by the electron beam onto the sample. Considering the above factors, our current characterization techniques are unable to achieve the goal of observing defects induced by strain in the Li metal-based materials. Despite this, extensive researches on metal cold deformation have demonstrated that metals generate numerous dislocations and point defects during cold deformation, and these defects with high energy are more likely to induce atomic diffusion.¹²⁻¹⁵ This can serve as the theoretical basis for establishing our hypothesis.

Comment 7. *In Figure 4b, the LACSS symmetric cell experienced a current density of 4mA/cm at the endpoints of CV scans without the sign of shorting. However, in Figure 4d, the LACSS symmetric cell showed immediate voltage drop when current density reached 2.5mA/cm². Can the author explain why LACSS cell could withstand 4 mA/cm² during CV scans?*

Response: Thanks for your thoughtful comments!

We suggest that the disparity in the results observed between CV and CCD tests stems from their distinct testing conditions (Li stripping/plating capacity).

Firstly, we would like to emphasize that the measurement of CCD is strongly influenced by the capacity of each cycle during the testing process and the cumulative capacity throughout the entire process. For instance, past studies have shown that when using fixed-capacity (0.1-0.5 mAh cm⁻²) CCD testing methods, the measured CCD value is often much higher than that obtained from the fixed-time (1 h) CCD testing method.¹⁶ Therefore, solely considering the magnitude of current that a battery can withstand while disregarding cycling capacity and cumulative capacity is unreasonable.

Our CCD testing method utilizes the commonly employed approach of fixed charging and discharging time of 1 h. With increasing current, the capacity of each cycle also increases. Prior to the symmetric cell using LACSS experiencing a short-circuit at 2.5 mA cm⁻², it completed a full discharging/charging cycle at 2.4 mA cm⁻² with a capacity of 2.4 mAh cm⁻². Moreover, its cumulative plated/stripped capacity reached 30 mAh cm⁻². In this scenario, significant deteriorations in the electrode/electrolyte interface have occurred before the current density reached 2.5 mA cm⁻².

In contrast, during the CV tests, the potential is constrained within the range of -0.1 to 0.1 V, with a scan rate of 0.2 mV s⁻¹. Consequently, although the response current density of the LACSS symmetric cell in CV tests may exceed 2.5 mA cm⁻², the duration of this high current density is less than 100 seconds, implying a very limited plating/stripping capacity (< 0.1 mAh cm⁻²). Compared to our fixed-time CCD tests, the cycling and cumulative capacity during CV tests are much smaller. This can explain why the LACSS symmetric cell can withstand a high

current density of 4 mA cm^{-2} for a few seconds in the CV tests.

Additionally, the variation in the maximum current density a cell can endure due to cycling and cumulative capacity is evident not only in symmetric cells utilizing LACSS but also in symmetric cells assembled with bare Li (Fig. R3.14a-d, also referred to as Fig. 4a-d in the manuscript). As illustrated in Fig. R3.14a and R3.14c (also referred to as Fig. 4a and 4c in the manuscript), symmetric cells assembled with bare lithium display a CCD value of 1.0 mA cm^{-2} . Nevertheless, during CV testing, they can withstand a significantly higher current density of $> 3 \text{ mA cm}^{-2}$.

Fig. R3.14 Electrochemical performance of symmetric cells with bare Li and the LACSS. **a, b** CV curves of Li symmetric cell (**a**), and LACSS symmetric cell (**b**). **c, d** Voltage profiles of Li symmetric cell (**c**) and LACSS symmetric cell (**d**) during step-increased current density tests.

Comment 8. *In Figure 4c, the symmetric cells using bare Li metal shorted immediately when the current density reached 1 mA/cm^2 , but the symmetric cells using bare Li metal was still able to be cycled for 10h in Figure 4g with the same current density. Where is such discrepancy coming from?*

Response: Thank you for your thoughtful comments!

We propose such discrepancy may come from the following two factors. Firstly, during the CCD testing process, the Li symmetric cell using LPSC has actually undergone cycling for 18 h at current densities ranging from 0.1 mA cm^{-2} to 0.9 mA cm^{-2} (with a step increased current density of 0.1 mA cm^{-2}). This extended process has resulted in significant damage to the electrode/electrolyte interface. Secondly, due to inherent variability among individual batteries, it is reasonable to expect some fluctuation in performance within a certain range. Therefore, we assembled two additional Li symmetric cells using LPSC for CCD tests. As **Fig. R3.15**

illustrates, the CCD of the symmetric cell using LPSC is not a constant value of 1.0 mA cm^{-2} , but fluctuates within a certain range. The CCD value of one symmetric cell using LPSC can reach up to 1.2 mA cm^{-2} . Hence, we contend that it is reasonable to expect the bare Li symmetric cell depicted in Fig. 4g to cycle for 10 h at 1.0 mA cm^{-2} .

Fig. R3.15 Voltage profiles of other two batches of bare Li symmetric cell during galvanostatic discharging/charging tests at step-increased current densities. The CCD values are determined to be 0.9 mA cm^{-2} for the batch 2 and 1.2 mA cm^{-2} for the batch 3.

Comment 9. For the XPS results on the cycled cells, are Figure S15 and S16 collected on the surface of negative electrodes or the SSE pellets? How did the authors separate the SSE pellets from anodes with structural integrity after cycling? Why would the shoulder peak at low energy in Figure S16a be identified as “Li-Al-S” species when the Al signature is almost absent (Figure 16c)?

Response: Thank you for your professional comments!

The XPS results in **Fig. S15-16** (denoted as **Fig. R3.16-3.17**, also referred to as **Fig. S24-25** in the SI) are collected on the surface of negative electrodes. We utilized surgical tweezers to peel off the negative electrode and electrolyte, obtaining samples with exposed electrodes surface (without noticeable electrolyte powder adhesion) for XPS characterization. However, the success rate of this process is not particularly high. Consequently, a negative electrode sample that can be used for final XPS characterization often requires the disassembly of multiple symmetric cells cycled under the same conditions to obtain.

Also, we apologize for the misleading identification of the shoulder peak in **Fig. R3.17**. Based on the S 2p spectra of the Li/LPSC and LACSS/LPSC interface, compared to the peak position of Li₂S, the peak corresponding to the decomposition products of SSEs on the surface of LACSS exhibits a shift towards higher binding energies, indicating the generation of small amounts of substances in addition to Li₂S. This phenomenon is also observed in the P 2p spectra of the Li/LPSC and LACSS/LPSC interface. In addition, in the Al 2p spectrum, a peak also appears on the surface of LACSS at a binding energy of 75 eV, corresponding to the presence of Al³⁺ rather than metallic Al. Hence, it is inferred that there is a minor reaction between the Li₉Al₄ alloy on the surface of LACSS and the LPSC electrolyte.

Therefore, in our revisions, we propose that this shoulder peak should be identified as the “Al-S/Li₂S” rather than “Li-Al-S” in **Fig. R3.17a**, the shoulder peak at low energy in **Fig. R3.17b** should be “Al-P/Reduced P”, and the peak in **Fig. R3.17c** should be “Al-P/Al-S”.

Fig. R3.16 XPS spectra of the bare Li/LPSC interface after cycling. **a** S 2p spectra. **b** P 2p spectra.

Fig. R3.17 XPS spectra of the LACSS/LPSC interface after cycling. **a** S 2p. **b** P 2p. **c** Al 2p.

The corresponding revisions in our paper are listed as follows:

-Manuscript, Results and Discussion, Stabilization of the Li/SSE interface by the LACSS

"...As illustrated by the XPS results of the LACSS/LPSC interface (Fig. S25), only a minor quantity of reduction products formed through the reaction between Li_9Al_4 and LPSC was observed."

Comment 10. The authors claimed that "...an extremely high average Young's modulus of 64 GPa, which plays a significant role in depressing potential Li penetration..." However, Li penetration is only harmful when it happens within SSE layer. Li penetration does not have much to do with the anode itself.

Response: Thank you for your valuable comments!

As addressed in our response to your **Comment 1** and **Comment 2**, Li deposition occurs underneath the outermost layer of the LACSS, which consists of only Li_9Al_4 and LiCl phases. Therefore, in our study, we propose that the LACSS presents a SEI/negative electrode integrated feature. Considering this, the Young's modulus of the LACSS surface characterized by AFM precisely reflects the mechanical properties of the SEI in the LACSS. According to previous studies,^{17, 18, 19} an increase in the Young's modulus of the SEI effectively suppresses dendrite penetration both in liquid-state and solid-state lithium metal batteries. The explanation is that when lithium deposits beneath the SEI, it first needs to penetrate the SEI before it can proceed to penetrate the solid electrolyte or membrane.

Comment 11. *What is the full cell capacity in mAh/cm^2 ? What is the N/P ratio? The deposition of Li metal for only $0.5 \text{ mAh}/\text{cm}^2$.*

Response: Thank you for your valuable comments!

For the long-term cycling stability tests of ASSLMBs, the theoretical areal capacity of active materials in the positive electrode is 1.78 mAh cm^{-2} and the current density for 0.5 C is 0.89 mA cm^{-2} . The areal capacity of the LACSS negative electrode is 11 mAh cm^{-2} . Therefore, the theoretical N/P ratio of the LACSS|LPSC|NCM83125 full cell used in the long-term cycling

stability tests is 6.18. As for the rate tests, the theoretical areal capacity of active materials in the positive electrode is 1.07 mAh cm⁻² and the areal capacity of the LACSS negative electrode is 11 mAh cm⁻². The theoretical N/P ratio of the LACSS|LPSC|NCM83125 full cell in the rate tests is 10.28.

The corresponding revisions in our paper are listed as follows:

-Manuscript, Methods, Cell assembly and electrochemical measurements

"The areal capacity of the 60 μm thick LACSS was measured to be 11 mAh cm⁻² by discharging the LACSS||Cu half cell with a liquid electrolyte (1 M LiPF₆ in EC:DEC (1:1 in volume) with 5 wt% FEC) to the cutoff voltage of -0.5 V (Fig. S30). The areal capacity of the 100 μm thick bare Li was calculated to be 20 mAh cm⁻². For the full cells for long-term cycling performance tests, the cathode loading was 1.78 mAh cm⁻². The theoretical N/P ratios of the LACSS|LPSC|NCM83125 full cell and the Li|LPSC|NCM83125 used in the long-term cycling stability tests were 6.18 and 11.23, respectively. For the full cells for rate and GITT tests, the cathode loading was 1.07 mAh cm⁻². The theoretical N/P ratios of the LACSS|LPSC|NCM83125 full cell and the Li|LPSC|NCM83125 used in rate performance and GITT tests were 10.28 and 18.69, respectively."

Comment 12. *Why did the full cell need activation process? What was occurring in the cell during the activation cycles? Was there any external pressure applied during all the electrochemical tests?*

Response: Thank you for your valuable comments!

Firstly, we would like to address the concerns on "activation process". When operating the ASSLBs directly at a high rate, it is common to observe a phenomenon that the capacity increases during the first initial cycles, a trend reported in numerous studies.²⁰⁻²⁴ A possible explanation is that when operating the battery at a high current density, partial active material may not participate in the initial electrochemical process. As the cycling progresses, these inactive cathode materials gradually participate in the electrochemical process, leading to a gradual increase in the battery's capacity. In our previous manuscript, we attribute this to the

activation of the cathode. We did not perform additional conventional activation of the cells. However, when addressing your concerns, we realized that the term of “activation process” may be misleading. **Therefore, we have made the following revisions to the manuscript.**

-Manuscript, Results and Discussion, Performance of ASSLMBs using the LACSS

“..As shown in Fig. 6a, the LACSS|LPSC|NCM83125 full cell delivers a maximum areal capacity of 1.30 mAh cm⁻² at 0.89 mA cm⁻². After 300 cycles, it maintains a reversible areal capacity of 1.20 mAh cm⁻², with a capacity retention of 92.6%. This remarkable performance is primarily attributed to the enhanced Li/SSE interfacial stability, as indicated by the consistently low voltage polarization throughout the cycles of the full cell using the LACSS (Fig. 6b). In contrast, when the LACSS was replaced by bare Li, the full cell experienced substantial capacity decay due to severe voltage polarization and encountered a short-circuit at the 80th cycle (Fig. 6c).”

Secondly, we did apply external pressure during the electrochemical tests of full cells. As **Fig. R3.18** (also referred to as **Fig. S29** in the SI) illustrates, the external force on the full cell is 49 Kg. Based on this, the pressure applied on the full cell is calculated to be 6 MPa.

Fig. R3.18 Measured force on the testing full cell.

The corresponding revisions in our paper are listed as follows:

-Manuscript, Methods, Cell assembly and electrochemical measurements

“..All the full cells were tested under a pressure of 6 MPa (Fig. S29).”

Some minor ones:

Comment 1. *The XPS peak fitting in Figure S3 is incorrect. Al peaks and AlCl peaks should not be totally covered under the other species.*

Response: Thank you for your valuable suggestion!

In numerous previous reports,^{25, 26, 27} Al and its compounds are often treated not as split double peaks but as a single peak in the Al 2*p* spectra. According to this, we have revised the XPS fitting and the result is presented in **Fig. R3.19** (also referred to as **Fig. S3** in the SI). In this way, Al peaks and AlCl₃ peaks are not totally covered under the other species. In addition, we also revised the **Fig. R3.20** (also referred to as **Fig. 2h** in the manuscript) based on this logic.

Fig. R3.19 Compositional evolutions after the deposition of sublimated AlCl₃ on Li metal. **a, b** Al 2*p* (**a**) and Cl 2*p* (**b**) XPS spectra of pristine AlCl₃ powder and the AlCl₃ deposited on Li.

Fig. R3.20 In-depth XPS profiles of Al 2p and Cl 2p spectra of the LACSS.

Comment 2. Equation 1 needs to list Li metal as the reactant and be balanced.

Response: Thank you for your thorough review!

We have list Li metal as the reactant and balance the equation 1.

The corresponding revision in our manuscript is listed as follow:

Comment 3. The schematic in Figure 3a is a bit confusing as it seems to indicate LACSS is sandwiched between Li_9Al_4 and LiCl phases.

Response: Thank you for your valuable comments!

We have revised the **Fig. 3a**. The new **Fig. 3a** (denoted as **Fig. R3.21**) is presented below.

Fig. R3.21 Thermodynamically unfavorable Li/ Li_9Al_4 and Li/LiCl interfaces.

Comment 4. *The overall language could use some major improvement.*

Response: Thank you for your valuable suggestion!

We have polished the language of the whole manuscript for better readership.

Reference

1. Fu L, *et al.* A lithium metal anode surviving battery cycling above 200 degrees C. *Adv Mater* **32**, e2000952 (2020).
2. Yang Lu, *et al.* The void formation behaviors in working solid-state Li metal batteries. *Sci Adv* **8**, eadd0510 (2022).
3. Peng Shi, *et al.* Inhibiting intercrystalline reactions of anode with electrolytes for long-cycling lithium batteries. *Sci Adv* **8**, eabq3445 (2022).
4. Liu Y, *et al.* Revealing the Impact of Cl substitution on the crystallization behavior and interfacial stability of superionic lithium argyrodites. *Adv Func Mater* **32**, 2207978 (2022).
5. Paul A, Laurila T, Divinski S. Chapter 1 - Defects, Driving Forces and Definitions of Diffusion Coefficients in Solids. In: *Handbook of Solid State Diffusion, Volume 1* (eds Paul A, Divinski S). Elsevier (2017).
6. Liang X, *et al.* A facile surface chemistry route to a stabilized lithium metal anode. *Nat Energy* **2**, 17119 (2017).
7. Wan HL, *et al.* Critical interphase overpotential as a lithium dendrite-suppression criterion for all-solid-state lithium battery design. *Nat Energy* **8**, 473-481 (2023).
8. Wan HL, Wang ZY, Zhang WR, He XZ, Wang CS. Interface design for all-solid-state lithium batteries. *Nature* **623**, 739-744 (2023).
9. Zhong XL, Haigh SJ, Zhou X, Withers PJ. An in-situ method for protecting internal cracks/pores from ion beam damage and reducing curtaining for TEM sample preparation using FIB. *Ultramicroscopy* **219**, 113135 (2020).
10. Yang H, Lozano JG, Pennycook TJ, Jones L, Hirsch PB, Nellist PD. Imaging screw dislocations at atomic resolution by aberration-corrected electron optical sectioning. *Nat Commun* **6**, 7266 (2015).
11. Chu S, *et al.* In situ atomic-scale observation of dislocation climb and grain boundary evolution in nanostructured metal. *Nat Commun* **13**, 4151 (2022).
12. Lima MNdS, *et al.* Influence of cold deformation on microstructure, crystallographic orientation and tensile properties of an experimental austenitic Fe–26Mn-0.4C steel. *J Mater Res Technol* **19**, 7-19 (2022).

13. Feng Z, Zecevic M, Knezevic M. Stress-assisted ($\gamma \rightarrow \alpha'$) and strain-induced ($\gamma \rightarrow \epsilon \rightarrow \alpha'$) phase transformation kinetics laws implemented in a crystal plasticity model for predicting strain path sensitive deformation of austenitic steels. *Int J Plast* **136**, 102807 (2021).
14. Agnihotri OP. Dislocation densities in heavily cold-worked copper and aluminium. *Br J Appl Phys* **17**, 603 (1966).
15. Yamaguchi Y, *et al.* Atomic diffusion of indium through threading dislocations in InGaN quantum wells. *Nano Lett* **22**, 6930-6935 (2022).
16. Wang TR, *et al.* A self-regulated gradient interphase for dendrite-free solid-state Li batteries. *Energy Environ Sci* **15**, 1325-1333 (2022).
17. Gao Y, *et al.* Unraveling the mechanical origin of stable solid electrolyte interphase. *Joule* **5**, 1860-1872 (2021).
18. Su H, *et al.* Stabilizing the interphase between Li and argyrodite electrolyte through synergistic phosphating process for all-solid-state lithium batteries. *Nano Energy* **96**, 107104 (2022).
19. Tsai W-Y, Thundat T, Nanda J. Toward a mechanically stable solid electrolyte interphase. *Matter* **4**, 2119-2122 (2021).
20. Wang Y, *et al.* Stable Ni-rich layered oxide cathode for sulfide-based all-solid-state lithium battery. *eScience* **2**, 537-545 (2022).
21. Kim JT, *et al.* An argyrodite sulfide coated NCM cathode for improved interfacial contact in normal-pressure operational all-solid-state batteries. *J Mater Chem A* **11**, 20549-20558 (2023).
22. Zhang Z, *et al.* An ultraconformal chemo-mechanical stable cathode interface for high-performance all-solid-state batteries at wide temperatures. *Energy Environ Sci* **16**, 4453-4463 (2023).
23. Shi B-X, *et al.* Mitigating contact loss in Li₆PS₅Cl-based solid-state batteries using a thin cationic polymer coating on NCM. *Adv Energy Mater* **13**, 2300310 (2023).
24. Lu P, *et al.* Superior low-temperature all-solid-state battery enabled by high-ionic-conductivity and low-energy-barrier interface. *ACS Nano* **18**, 7334-7345 (2024).
25. Adhitama E, *et al.* Revealing the Role, Mechanism, and impact of AlF₃ coatings on the

- interphase of silicon thin film anodes. *Adv Energy Mater* **12**, 2201859 (2022).
26. Li D, Chu F, He Z, Cheng Y, Wu F. Single-material aluminum foil as anodes enabling high-performance lithium-ion batteries: The roles of prelithiation and working mechanism. *Mater Today* **58**, 80-90 (2022).
27. Qin B, *et al.* Revisiting the electrochemical lithiation mechanism of aluminum and the role of Li-rich phases (Li_{1+x}Al) on capacity fading. *ChemSusChem* **12**, 2609-2619 (2019).

REVIEWERS' COMMENTS

Reviewer #1 (Remarks to the Author):

Authors have addressed all my concerns in the revised manuscript. I thus recommend publication of the manuscript in its present form.

Reviewer #2 (Remarks to the Author):

The authors have addressed all the comments satisfactorily. the manuscript is now ready to be published in Nature Communications.

Reviewer #3 (Remarks to the Author):

The authors have addressed my questions.